# Human immunocompetent Organ-on-Chip platforms allow safety profiling of tumor-targeted T-cell bispecific antibodies

S Jordan Kerns[1†], Chaitra Belgur[1†], Debora Petropolis[1], Marianne Kanellias[1], Riccardo Barrile[1,2], Johannes Sam[3], Tina Weinzierl[3], Tanja Fauti[3], Anne Freimoser-Grundschober[3], Jan Eckmann[4], Carina Hage[4], Martina Geiger[3], Patrick Ray Ng[1], William Tien-Street[1], Dimitris V Manatakis[1], Virginie Micallef[5], Regine Gerard[5], Michael Bscheider[5], Ekaterina Breous-Nystrom[5], Anneliese Schneider[3], Anna Maria Giusti[3], Cristina Bertinetti-Lapatki[5], Heather Shannon Grant[1], Adrian B Roth[5], Geraldine A Hamilton[1], Thomas Singer[5], Katia Karalis[1], Annie Moisan[5], Peter Bruenker[3], Christian Klein[3], Marina Bacac[3], Nikolce Gjorevski[5‡*], Lauriane Cabon[5‡*]

[1]Emulate Inc, Boston, United States; [2]Department of Biomedical Engineering, University of Cincinnati, Cincinnati, United States; [3]Roche Pharma Research & Early Development, Roche Innovation Center Zurich, Schlieren, Switzerland; [4]Roche Pharma Research & Early Development, Roche Innovation Center Munich, Penzberg, Germany; [5]Roche Pharma Research & Early Development, Roche Innovation Center Basel, Basel, Switzerland

*For correspondence:
nikolche.gjorevski@roche.com (NG);
lauriane.cabon@roche.com (LC)

[†]These authors contributed equally to this work
[‡]These authors also contributed equally to this work

**Abstract** Traditional drug safety assessment often fails to predict complications in humans, especially when the drug targets the immune system. Here, we show the unprecedented capability of two human Organs-on-Chips to evaluate the safety profile of T-cell bispecific antibodies (TCBs) targeting tumor antigens. Although promising for cancer immunotherapy, TCBs are associated with an on-target, off-tumor risk due to low levels of expression of tumor antigens in healthy tissues. We leveraged in vivo target expression and toxicity data of TCBs targeting folate receptor 1 (FOLR1) or carcinoembryonic antigen (CEA) to design and validate human immunocompetent Organs-on-Chips safety platforms. We discovered that the Lung-Chip and Intestine-Chip could reproduce and predict target-dependent TCB safety liabilities, based on sensitivity to key determinants thereof, such as target expression and antibody affinity. These novel tools broaden the research options available for mechanistic understandings of engineered therapeutic antibodies and assessing safety in tissues susceptible to adverse events.

## Introduction

Cancer immunotherapy has received intense attention over the past two decades owing to the promise of delivering durable cures by harnessing the cytotoxic potential of the immune system against tumor cells (*Waldman et al., 2020*; *Yang, 2015*; *Gong et al., 2018*). However, although impressive improvement in long-term survival has been reported (*Hodi et al., 2010*; *Schadendorf et al., 2015*; *Wolchok et al., 2017*), only a fraction of patients responds. Furthermore, the systemic immunomodulation mediated by these drugs often elicits immune-related adverse events (irAEs), including skin and liver toxicity, colitis, and pneumonitis, limiting their broad clinical application in battling cancer (*Champiat et al., 2017*; *Naidoo et al., 2015*; *Kennedy and Salama, 2020*).

T-cell engaging bispecific antibodies (TCBs) are a novel class of cancer immunotherapeutic agents that have the potential to improve on the clinical efficacy and safety of traditional immunotherapy (*Clynes and Desjarlais, 2019*; *Labrijn et al., 2019*). TCBs exert their anti-tumor activity by simultaneously binding to a cancer surface antigen and the CD3 T-cell receptor, thereby both activating the latter and physically crosslinking it to target cells (*Bacac et al., 2016a*). This synthetic immunity approach is particularly favorable for targeting less immunogenic, neo-antigen-lacking tumors, as T cells can be recruited and activated independently of their T-cell receptor specificity. This strictly tumor-targeted immunomodulation is also expected to reduce the systemic inflammatory toxicities associated with traditional immunotherapies. The therapeutic potential of TCBs is exemplified by the large number of molecules targeting solid and blood tumors, which are currently in various stages of clinical evaluation (*Ishiguro et al., 2017*; *Goebeler and Bargou, 2020*).

Although TCBs hold the promise for a safer therapeutic option, they are not risk free. The antigens targeted are rarely exclusive to the tumor, but are also often expressed, albeit at lower levels, in normal tissues, rendering TCBs subject to 'on-target, off-tumor' safety liabilities. This is particularly true for epithelial tumor antigens as they are frequently targeted in solid tumor indications. For example, a Bispecific T-cell Engager (BiTE) targeted to the epidermal growth factor receptor (EGFR) produced severe liver and kidney toxicities in non-human primates, in line with EGFR expression in these organs, and led to the termination of the animals (*Klinger et al., 2016*; *Lutterbuese et al., 2010*). Clinical adverse events were reported in a recent Phase I study evaluating an epithelial cell adhesion molecule (EpCAM)-targeted BiTE as a therapy for a variety of epithelial carcinomas. Consistent with the expression of EpCAM in the gastrointestinal tract, the molecule triggered severe diarrhea and ultimately prevented escalation to efficacious doses and the identification of a therapeutic window (*Kebenko et al., 2018*; *Trabolsi et al., 2019*). Reliable human TCB safety evaluations at the preclinical stage are therefore of vital importance to ensure that well-tolerated and efficacious therapeutics reach patients.

Traditional rodent-based preclinical models are often ill-suited for predicting some cancer immunotherapy-mediated adverse events in humans in part because of the fundamental differences in the immunological responses between the species (*Bjornson-Hooper et al., 2019*). In the EpCAM example mentioned above, the severity of the diarrhea elicited by the treatment was not predicted by preclinical studies in mice (*Brischwein et al., 2006*). Moreover, an increasing number of TCBs target human-specific antigens that lack expression in animals, rendering preclinical animal studies uninformative for safety and efficacy assessments (*Bacac et al., 2016a*). Indeed, the development of preclinical models that better translate to human immunity is regarded as one of the top current challenges of cancer immunotherapy (*Hegde and Chen, 2020*).

While human-relevant cell-based models of tissues and organs promise to bridge this gap, conventional in vitro two-dimensional (2D) models fail to provide the complexity required to model the biological mechanisms of immunotherapeutic effects. Furthermore, their reductive microenvironment, devoid of essential cellular, biochemical, and biophysical factors found in the native organ, precludes the expression of TCB targets at physiologically relevant levels and patterns, crucial for capturing TCB pharmacology and safety liabilities.

Organ-on-Chip models aim to overcome these limitations by combining micro-engineering with cultured primary human cells to recreate the complex multifactorial microenvironment and functions of native tissues and organs (*Huh et al., 2010*). The tissue microenvironment in vivo provides the external signals that help drive cellular differentiation toward mature phenotypes. Organs-on-Chips model key functional aspects of tissue-level physiology such as epithelial and microvascular tissue-tissue interfaces, and physiologically relevant mechanical forces, have been shown to more accurately capture in vivo-relevant phenotypes (*Kasendra et al., 2020*; *Kasendra et al., 2018*; *Gayer and Basson, 2009*). The enhanced tissue maturation promoted by Organs-on-Chips could help ensure organ-specific expression of TCB targets, while the modularity of these devices and the possibility for controlled circulation of molecules and immune cells could better capture the functional interactions between TCBs, immune cells, and target-expressing cells that occur in patients. Motivated by these advantages, we set out to evaluate Organs-on-Chips as platforms for the assessment of on-target, off-tumor TCB safety risks in human organs, using a panel of targeting and non-targeting molecules, and leveraging in vivo target expression and toxicity data. We found that these systems could reproduce and predict target-dependent TCB safety liabilities, showing sensitivity to key determinants thereof, such as target expression and antibody affinity.

# Results

As a starting point for our method development and validation, we used molecules under current preclinical development. We focused on a T-cell bispecific antibody generated to bring Folate Receptor 1 (FOLR1) expressing tumor cells in close proximity to CD3 expressing cytotoxic T-cells (*Figure 1—figure supplement 1A*; *Geiger et al., 2020*). FOLR1 is overexpressed in many solid, epithelial-derived tumors including ovarian, lung, and breast cancer (*Scaranti et al., 2020*), but is also expressed to a lower degree on normal epithelial cells as found in the lung and kidneys (*Parker et al., 2005*). While a high-affinity FOLR1-TCB (FOLR1(Hi) TCB) was efficacious in human breast cancer patient-derived xenograft models (*Figure 1—figure supplement 1B*), severe on-target toxicity in the lung of cynomolgus monkey was observed (*AMea, 2016*). Clinical signs of severe respiratory inflammation appeared as early as 24 hr post-dosing, and pro-inflammatory cytokines IL-6, IL-2, and IL-8 were elevated in the blood of affected animals, and importantly coincided with an increase of inflammation markers. Histopathology assessment revealed leukocytic infiltrates in lung tissue indicative of immune-mediated toxicity (*Figure 1A*). Further immunohistochemistry (IHC) studies (*Figure 1B–D*) indicated low relative expression (compared to high FOLR1 expression in the ovarian carcinoma cell line, HeLa) of the FOLR1 target antigen in lung alveolar epithelial cells of cynomolgus lung tissue suggesting an adverse event largely driven by on-target toxicity. Importantly, IHC analysis of FOLR1 expression in the human lung revealed similar levels and patterns of the antigen as in the cynomolgus (*Figure 1E,F*), extending the threat of safety liabilities to patients in the clinical setting. Bearing in mind the toxicological profile of FOLR1(Hi) TCB in cynomolgus and the expression of FOLR1 in both species lung tissue, we identified the lung as an at-risk organ in patients and accordingly set out to evaluate a human Alveolus Lung-Chip model as a platform for FOLR1(Hi) TCB toxicology assessment.

Alveolus Lung-Chips were seeded with human adult lung primary alveolar epithelial cells on top of an extra-cellular matrix-coated porous membrane that separates two parallel, fluidic microchannels (*Figure 1G*, *Figure 2—figure supplement 1*). On the opposite side of the membrane, human primary lung microvascular cells were seeded to form a lower tubular vascular channel as described previously (*Jain et al., 2018*). A mature Alveolus Lung-Chip model was obtained after 5 days of liquid–liquid culture (LLI) followed by establishment of an air–liquid–culture (ALI) for a further 5 days. To evaluate FOLR1 expression in chips, we combined RNA sequencing, immunofluorescence and flow cytometry analyses. FOLR1 gene expression was even over time as shown by quantification of RNA transcripts (*Figure 1H*). We also confirmed FOLR1 protein expression in mature chips (*Figure 1I*). Flow cytometry-mediated quantification allowed us to estimate the cell surface expression of FOLR1 within the Alveolus Lung-Chip at an average of ~10,000 molecules expressed per cell (*Figure 1J*). For comparison, the high FOLR1 expressing ovarian carcinoma HeLa cell line displayed an average of ~450,000 molecules per cell when cultured on chip (*Figure 2—figure supplement 1B*), confirming the difference observed in IHC between healthy and tumor cells.

To render the device immunocompetent and capable of simulating on-target TCB-mediated immunomodulation, we added peripheral mononuclear blood cells (PBMC) isolated from human whole blood to the epithelial channel in direct contact with the mature alveolar epithelium (*Figure 2—figure supplement 1*). Introduction of T cells is required to engage the CD3 arm of FOLR1(Hi) TCB thereby allowing its mode of action. The sequence of events we aimed to reproduce in chips are the crosslinking of T cells to the FOLR1 expressing target cells mediated by the TCB, subsequent T-cell activation and cytolytic synapse formation resulting in cytotoxic granules release (granzymes and perforin) and consequent target cell apoptosis. Early T-cell cytokine release (TNFα, IFNγ) should be followed by later cytokine release from epithelial cells and monocytes (IL-6, IL1β, IL-8) combined with strong physical attachment of immune cells to the FOLR1-expressing lung epithelium via the TCB. Thus, we selected and optimized experimental readouts that would enable us to monitor these steps in the Alveolar Lung-Chip.

*Figure 2A* shows representative brightfield and fluorescent images of the Alveolus Lung-Chip at 48 hr after the administration of immune cells. Compared to the chips treated with a non-targeting (NT) TCB control antibody, FOLR1(Hi) TCB treated chips presented an increased apoptosis of the alveolar epithelium (*Figure 2B*). Consistently, FOLR1(Hi) TCB treatment led to increased T-cell activation (as evidenced by CD69 upregulation in CD8$^+$ T cells) in the presence of target-expressing cells (*Figure 2C*), but not in PBMC only (*Figure 2C*, right panel). Supernatants collected from the

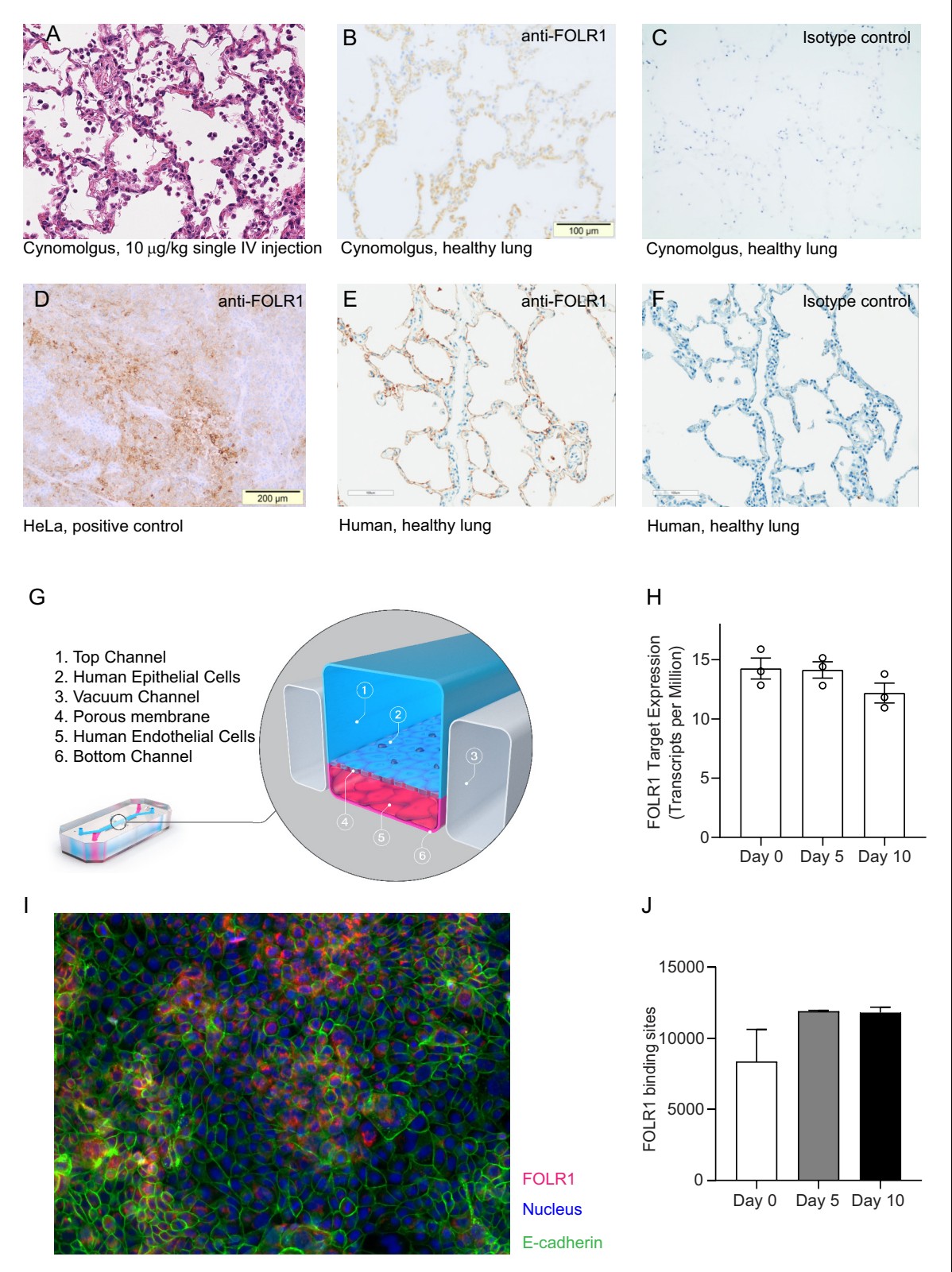

**Figure 1.** FOLR1 expression in the alveolar epithelium of cynomolgus and humans underlies on-target off-tumor toxicities of FOLR1-TCB and can be recreated in a human alveolus lung-chip. (**A**) IHC of pre-clinical, cynomolgus lung tissue 24 hr after intravenous single-dose administration of high-affinity FOLR1-TCB (FOLR1(Hi), 10 μg/kg), demonstrating leukocytic infiltration (dark purple cells) and inflammation. (**B**) Expression of FOLR1 protein in healthy cynomolgus lung tissue stained with antibody targeting FOLR1. (**C, F**) Isotype controls of FOLR1 staining in healthy cynomolgus and human

*Figure 1 continued on next page*

*Figure 1 continued*

lung tissues, respectively. (**D**) High FOLR1 expression displayed in human ovarian carcinoma HeLa cell line for comparison to (**E**) histopathological staining of primary healthy human lung tissue for FOLR1. (**G**) Schematic of Alveolus Lung-Chip to model human FOLR1 on-target toxicities. Alveolus Lung-Chip design is composed of a top microfluidic channel (*Waldman et al., 2020*) seeded with primary adult human alveolar cells (*Yang, 2015*) cultured to maturity with air–liquid interface (ALI). The top, epithelial channel is separated with a flexible, porous membrane (*Hodi et al., 2010*) from a bottom, vascular channel seeded with primary lung microvascular cells (*Schadendorf et al., 2015*; *Wolchok et al., 2017*). Mechanical stretching is applied via pneumatic actuation of parallel vacuum channels (*Gong et al., 2018*). (**H**) RNAseq expression levels of FOLR1 gene in cultured alveolar epithelial cells on day 0 (before seeding), 5, or 10 after seeding and differentiation on the Alveolus Lung-Chip (n=3, ± SEM). (**I**) Representative immunofluorescent staining of chip epithelium (Nuclei, blue) at day 10 of culture expressing the tight junction marker E-cadherin (green) and FOLR1 target antigen (red). Images taken at 40× magnification. (**J**) Estimation of surface FOLR1 binding site expression via flow cytometry of harvested chip epithelial cells at days 0 (before seeding), 5, and 10 (n=2–4, ± SEM).

The online version of this article includes the following figure supplement(s) for figure 1:

**Figure supplement 1.** Anti-tumor potency and efficacy of FOLR1-targeted TCBs.

outlet reservoir of the epithelial channels at 24 and 48 hr post-treatment were measured for multi-plex cytokines (*Figure 2D*), revealing a significantly increased secretion of IFNγ at 24 hr and 48 hr that correlated with increased granzyme B and IL-6 at 48 hr in response to FOLR1(Hi) TCB.

Interestingly, we noticed a higher number of immune cells in the FOLR1(Hi) TCB condition compared to the NT control, possibly due to a combination of T-cell proliferation and increased attachment of T cells (*Figure 2A*). Quantification of the immune cell presence in fixed chips confirmed increased attachment of both T cells and non CD3⁺ cells to the target epithelium in the FOLR1(Hi) TCB condition (*Figure 2E,F*), which is consistent with the TCB mode of action, whereby immune cells are crosslinked to target cells. Together, these data suggest that the Alveolus Lung-Chip successfully replicates aspects of the FOLR1 (Hi) TCB-mediated toxicity observed in cynomolgus, and suggests that the human lung would be subject to similar safety liabilities.

In light of the toxicity risk predicted above, and hoping to define a potential therapeutic window of FOLR1(Hi) TCB, we performed the same study with chips seeded with the high FOLR1 expressing ovarian carcinoma cell line, HeLa, previously used to assess drug efficacy (*Geiger et al., 2020*). Although no effects were seen at the lowest concentration, all concentrations starting from 2 ng/mL induced significant T-cell activation (measured at 48 hr), cancer cell apoptosis (from 24 hr onwards) and strong cytokine release, as expected from the high level of FOLR1 expression in HeLa cells (*Figure 2—figure supplement 2*). Thus, we efficiently killed tumor cells at a much lower concentration than that needed to induce damage to the healthy alveolar epithelial cells. Of note, FOLR1(Hi) TCB EC50 was estimated at 1.1 pM, which was close to the value obtained from standard 2D in vitro killing experiments (2.2 pM).

Although the data described above suggested that a therapeutic window for FOLR1(Hi) TCB could be determined, we leveraged the chip to instead identify a safer molecule. In particular, we utilized an antibody with lower monovalent affinity for the FOLR1 target, referred to as FOLR1(Lo) TCB making use of avidity mediated selectivity gain (*Figure 1—figure supplement 1C,D*). As a result of that design, FOLR1(Lo) TCB presented a lower binding to FOLR1-expressing HeLa cells while retaining a potent killing activity in coculture assays and an in vivo tumor control efficacy (*Figure 1—figure supplement 1E–G*). We profiled the two TCBs, FOLR1(Hi) and FOLR1 (Lo), in the immunocompetent Alveolus Lung-Chip following the workflow and readouts described above, and found that FOLR1(Hi) TCB induced a significant increase in all the readouts starting at the 0.2 μg/mL concentration whereas FOLR1(Lo) TCB showed a response only at the highest concentration of 20 μg/mL and to a much lower magnitude than the high-affinity molecule (*Figure 3A–D*, *Figure 3—figure supplement 1*). These results demonstrated that the cellular responses on the platform are sensitive to differences in antibody affinity and recapitulates the biology associated with the mode of action of TCBs. Following these in vitro observations, the lower affinity FOLR1 TCB was tested in a cynomolgus toxicology study and none of the animals experienced the lung inflammation observed with FOLR1(Hi) TCB (*Figure 3E*), confirming the safer profile predicted by the Alveolus Lung-Chip.

To compare the chip format to its 2D equivalent, we benchmarked the Alveolus Lung-Chip against transwell inserts coated with alveolar epithelial cells and endothelial cells on opposing sides (*Figure 3—figure supplement 2*). In the transwell environment, quantification of TCB-dependent immune cell attachment and apoptosis did not show differences between the control and FOLR1

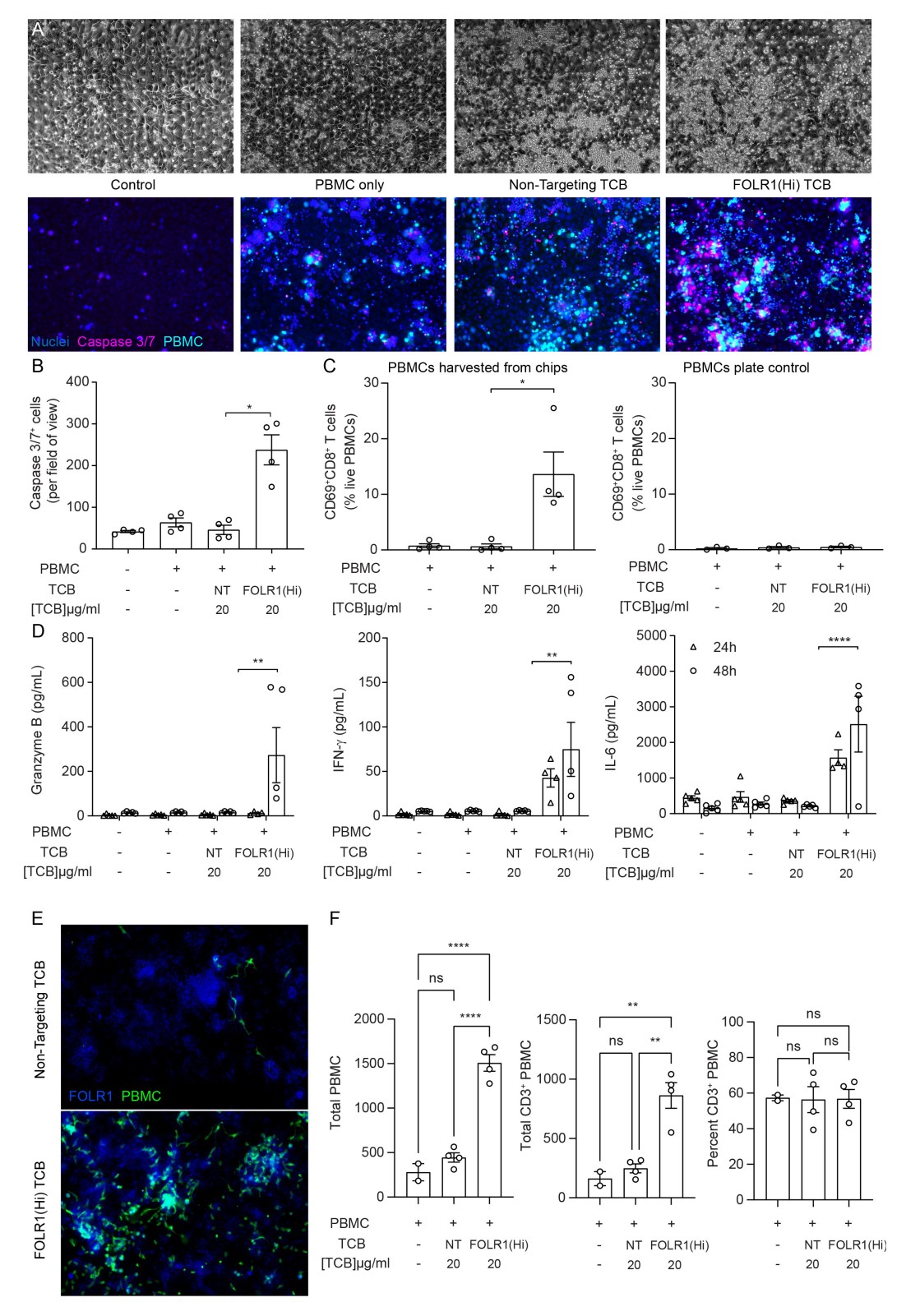

**Figure 2.** The immunocompetent Alveolus Lung-Chip recapitulates TCB-mediated on-target off-tumor toxicity. Isolated PBMCs were pre-incubated for 1 hr with high-affinity FOLR1 TCB (FOLR1(Hi)) or non-targeting TCB control (NT) and introduced to the epithelial channel of differentiated Alveolus Lung-Chips and rested for 3 hr prior to initiation of media perfusion. The established co-culture with immune cells was then maintained for 48 hr under flow with fresh media. (**A**) Representative brightfield (top) and immunofluorescent images (bottom) of Alveolar Lung-Chip epithelium (nuclei, blue) 48 hr

*Figure 2 continued on next page*

*Figure 2 continued*

after addition of PBMC (cyan). The control group did not have PBMC administered. The FOLR1(Hi) group showed higher levels of PBMC attachment and caspase-3/7-positive, apoptotic cells (magenta) (B) Quantification of apoptotic caspase-3/7-positive cells collected on live chips (n=4). (C) Flow cytometry analysis of PBMC harvested from chips or plates for percentage of live, CD69$^+$ activated CD8$^+$ T cells (n=4 approx. 10,000 cells per chip) after 48 hr (n=4). PBMC cultured on plates after 48 hr incubation showed overall low activation levels without attachment to epithelium. (D) Multiplex cytokine analysis of epithelial channel supernatants at 24 and 48 hr after PBMC introduction (n=4). (E) Immunofluorescent staining of FOLR1 target expression (blue) in epithelium of chips administered with NT control (left) and FOLR1(Hi)-treated (right) PBMC (green). Increased accumulation of PBMC and co-localization with FOLR1 signal was observed in FOLR1(Hi) group. (F) Quantification of immunofluorescent images confirmed increased PBMC attachment (including T cells) in the FOLR1(Hi) group (n=4). Statistical analysis was conducted by one-way ANOVA (B, C, D, F) and was defined as *p<0.05, **p<0.01, and ***p<0.001. Errors bars represent ± SEM.

The online version of this article includes the following figure supplement(s) for figure 2:

**Figure supplement 1.** Experimental outline of Alveolus-Chip model.

**Figure supplement 2.** HeLa Lung-Chip Produces On-Target T-cell Killing Response.

TCB treatment conditions. T-cell activation and T-cell-specific cytokines were elevated, possibly due to prolonged interactions of the immune cells with the epithelium and static media supply. Also, the transwell version did not capture differences between FOLR1 antibodies: FOLR1(Lo) TCB led to a higher amount of granzyme B and IFNγ release than the chips, which is inconsistent with the absence of toxicity observed in vivo. Given the higher concordance of the results produced by the Alveolar Lung-Chip compared with the transwell counterpart, we propose that the novel immunocompetent Alveolus Lung-Chip platform can faithfully evaluate TCB on-target, off-tumor risks and presents a superior value to the existing alternatives.

To demonstrate the broad applicability of the model for testing target-mediated TCB safety risks, we extended the methodology to a second target and a second Organ-Chip model. In this example, we focused on TCBs targeting carcinoembryonic antigen (CEA), which is overexpressed in a range of solid tumors, including colorectal cancer (*Hammarström, 1999*). We have created TCBs binding to CEA with high or low affinity – CEA(Hi) and CEA(Lo) TCB, respectively (*Figure 4—figure supplement 1*), and are currently evaluating them as therapies for a range of solid tumors. Indeed, we have found that CEA(Hi) and CEA(Lo) TCB are potent mediators of tumor cell lysis and T-cell activation in vitro (*Figure 4—figure supplement 1B,C*), and exhibit robust anti-tumor activity in humanized mice engrafted with CEA-expressing tumors (*Figure 4—figure supplement 1C*).

Aside from solid tumors, however, CEA is expressed in the gastrointestinal tract (*Benchimol et al., 1989*; *Thomas et al., 1995*; *Zhou et al., 1993*). Immunohistochemistry analysis of primary human intestinal samples confirmed high expression of CEA in the colon, whereas small intestinal expression was lower. In both tissues, CEA was enriched on the apical surface of the barrier (*Figure 4A*). The substantial target presence in the gastrointestinal tract implicates this system as an at-risk organ, motivating us to assess potential intestinal toxicities triggered by CEA-engaging TCBs. Our antibodies recognize a human-specific epitope within the CEA protein,, making mice and cynomolgus preclinical toxicology models unsuitable for the assessment of toxicities caused by CEA-targeting TCBs. Indeed, our antibodies showed lack of cross-reactivity to cynomolgus monkey CEA, which underscores the need for human-relevant models in addressing this question (*Figure 4—figure supplement 1E*). Therefore, we leveraged the recently developed and characterized Colon and Duodenum Intestine-Chips, which combine the two most advanced approaches in the field of intestinal modeling – primary human organoids and Organs-on-Chips (*Kasendra et al., 2020*; *Kasendra et al., 2018*). Briefly, primary human colon intestinal organoids are dissociated and seeded within an Organ-Chip (*Figure 4—figure supplements 2* and *3A*), where they form a tight, polarized barrier, containing the full range of mature intestinal cell types. We have previously shown that the inclusion of luminal flow, peristaltic motion, and an endothelial layer enhances the maturation and physiological fidelity of the barrier, compared with organoids in conventional 3D culture (*Apostolou et al., 2020*).

To qualify the Intestine Chip as a platform for TCB safety assessment, we set out to determine whether it (1) supports physiologically relevant target expression and (2) can successfully capture target-mediated TCB toxicity. Immunofluorescence analysis revealed robust expression of CEA in the Colon Intestine-Chip epithelium, whereas expression in the Duodenum Intestine-Chip appeared weaker and localized to the apical surface (*Figure 4B*).

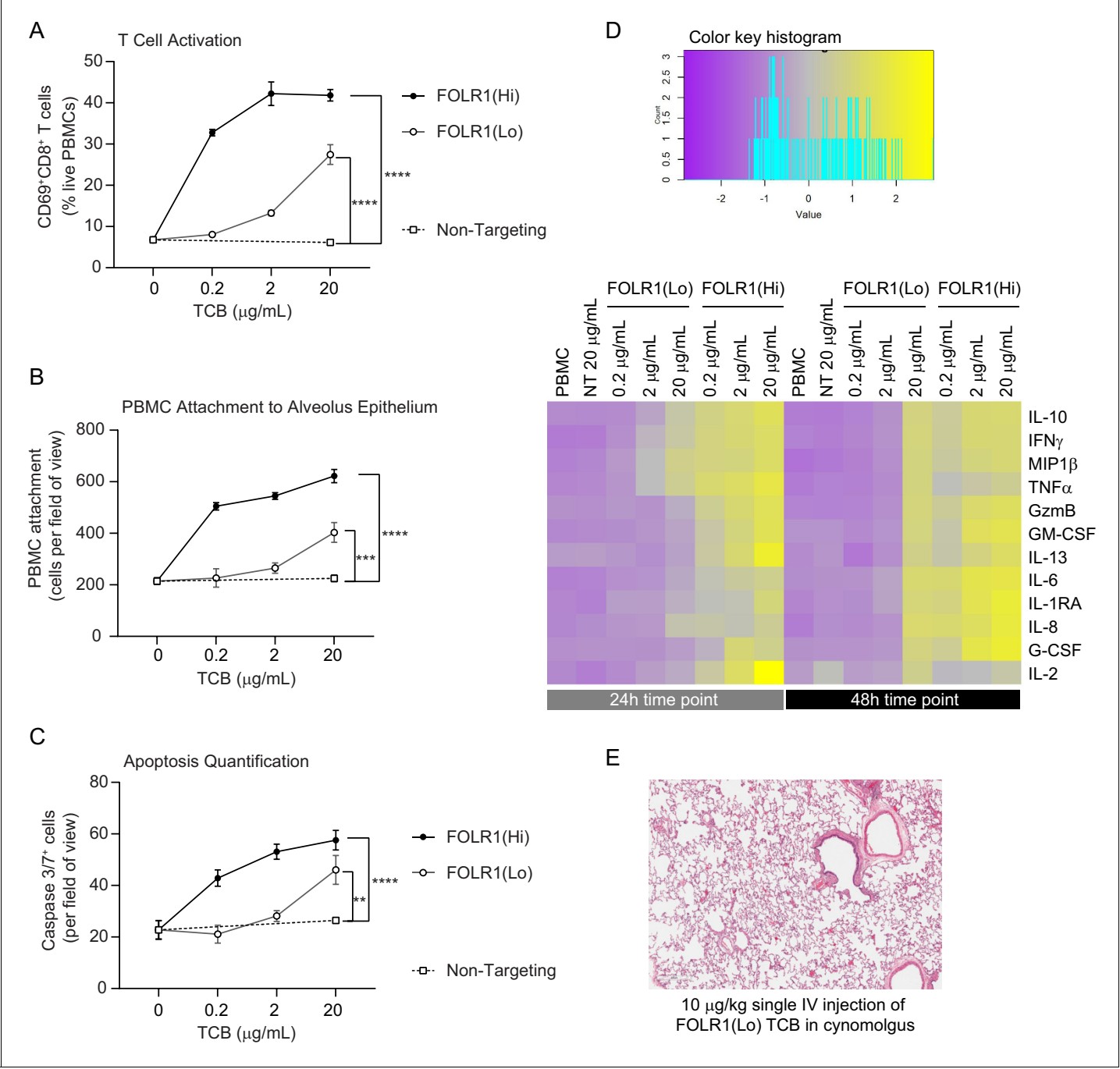

**Figure 3.** Dose–response and TCB affinity-dependent effects displayed in immunocompetent Alveolus Lung-Chips. (**A**) Flow cytometry analysis of PBMC harvested from epithelial channel and assessed for CD69+ activated CD8+ T cells (n=4 approx. 10,000 cells per chip). (**B**) Quantification of immunofluorescent images of prelabelled (PBMC that remained attached after harvest from chip epithelium) (n=4). (**C**) Immunofluorescent image quantification of caspase-3/7+ apoptotic epithelial cells at 48 hr time point. The NT control group displayed no increase in T-cell activation, PBMC attachment, or apoptotic cells with increasing dose, while the FOLR1(Lo) group showed a significant increase at 20 µg/mL and the FOLR1(Hi) group displayed an increase from 0.2 µg/mL in a dose-dependent manner (n=4). (**D**) Heat map displaying multiplex cytokine analysis of chip epithelial channel supernatant at 24 and 48 hr post-treatment. (**E**) Histological lung tissue section from pre-clinical cynomolgus study of intravenous FOLR1(Lo) (10 µg/kg), 24 hr after administration. Statistical analysis was conducted by one-way ANOVA (**A, B, C**) and was defined as **p<0.01, ***p<0.001, and ****p<0.0001. Errors bars represent ± SEM.

The online version of this article includes the following figure supplement(s) for figure 3:

**Figure supplement 1.** Proportions of T-cell subsets harvested from the Alveolus Lung-Chip platform.

**Figure supplement 2.** Comparison of T-cell killing response of transwell culture to alveolus lung-chip.

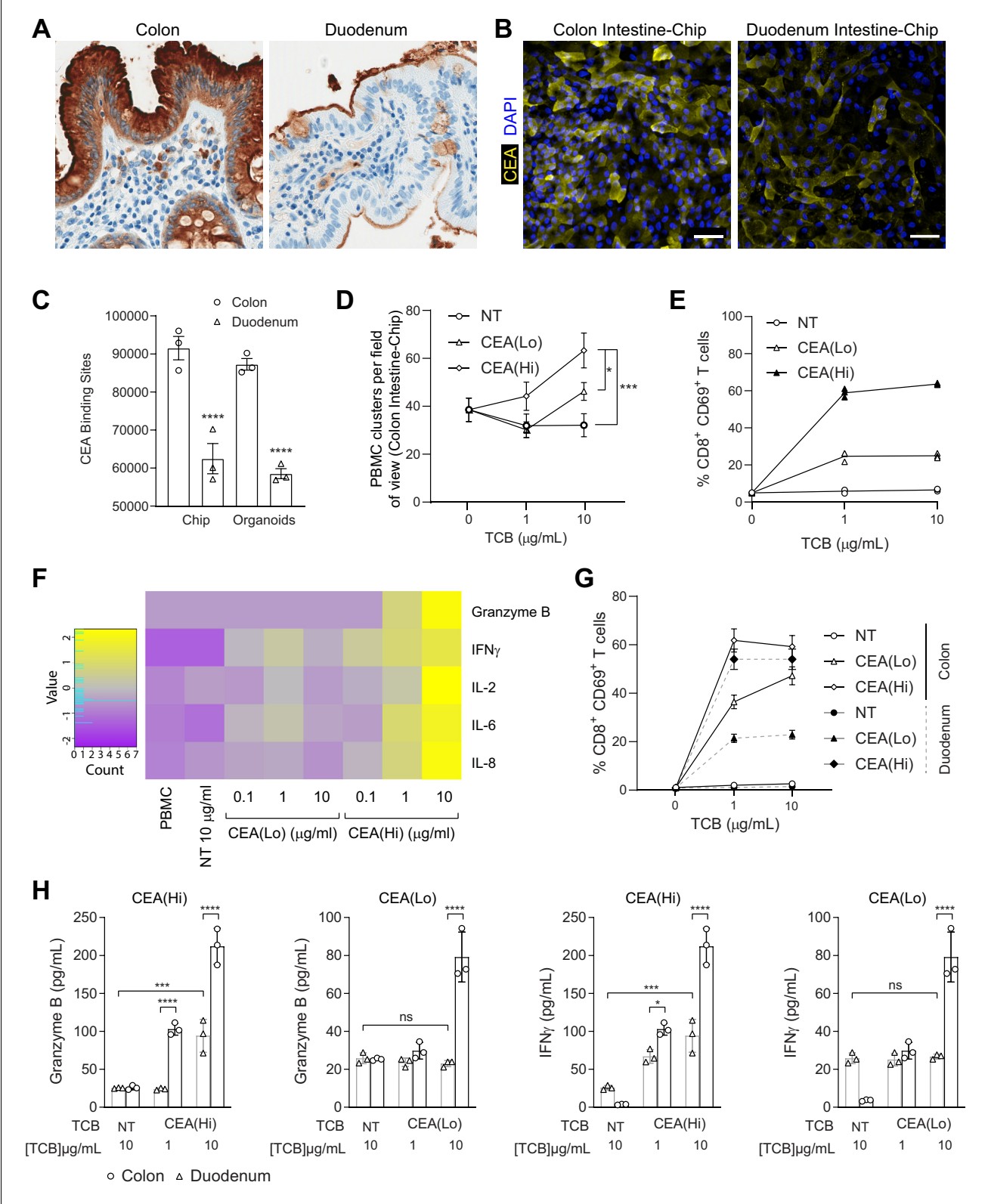

**Figure 4.** Application of colon intestine-Chip as model of CEA-TCB-mediated adverse effects. (**A**) IHC of human colon and duodenum tissue stained with anti-CEA (brown coloration) demonstrating difference in regional expression. (**B**) Representative immunofluorescent micrograph depicting CEA expression in the epithelial compartment of the Colon Intestine-Chip and Duodenum Intestine-Chip. (**C**) Average cell surface expression of CEA within 3D organoids and chips at day 8 of culture (n=3). (**D**) Colon-Chip epithelial channels were administered with PBMC treated with/without low and high-

*Figure 4 continued on next page*

*Figure 4 continued*

affinity (CEA(Lo) and CEA(Hi)) TCB (0.1–10 µg/mL), or Non-targeting (NT) TCB (10 µg/mL). Co-culture was maintained under flow for 72 hr. Quantification of immunofluorescent images collected live indicate multiple clusters of PBMC settled throughout epithelial structures. Statistical analysis was conducted by one-way ANOVA and was defined as *p<0.05 and ***p<0.001. Errors bars represent ± SEM. (E) CD69$^+$ Activation of CD8$^+$ T cells of harvested PBMC measured by flow-cytometry (n=3± SEM). (F) Heat map of multiplex cytokine panel from epithelial channel supernatants. Data (D–F) from terminal endpoint 72 hr after administration (n=3). (G) Colon- and Duodenum-Chips were administered with PBMC with low and high-affinity (CEA (Lo) and CEA(Hi)) TCB treatment from 0 to 10 µg/mL, along with Non-targeting (NT) control. Flow cytometry analysis of harvested PBMC from chips 72 hr post-treatment to measure levels of activated CD69$^+$CD8$^+$ T cells (n=3± SEM). (H) Multiplex cytokine analysis of supernatant collected from epithelial channels of Colon and Duodenum-Chips after 72 hr of treatment (n=3± SEM).

The online version of this article includes the following figure supplement(s) for figure 4:

**Figure supplement 1.** Anti-tumor potency and animal cross-reactivity of CEA-targeted TCBs.
**Figure supplement 2.** Experimental outline of Intestine-Chip model.
**Figure supplement 3.** Intestine-Chip CEA expression, comparison to conventional models and target-independent PBMC activation of CEA-targeted TCBs.

Flow cytometry-mediated quantification verified the cell surface expression of CEA within the Intestine-Chip and confirmed the higher target abundance in the Colon-Chip compared with the Duodenum-chip (*Figure 4C*), in line with the regional differences observed in vivo. Based on previous data suggesting that CEA-targeting TCBs trigger immune cell activation and cancer cell killing above a threshold of 10,000 CEA molecules (*Bacac et al., 2016b*), we expect to detect on-target TCB toxicity in the Intestine-Chip. To our surprise, colon and duodenum organoids cultured in the conventional 3D format featured target expression levels similar to those measured in the chips (*Figure 4C*). Nonetheless, despite the similarity in target abundance, we believe that the Intestine-Chip is more suitable for the assessment of on-target TCB toxicities, given the more physiologically accurate CEA expression patterns compared with those observed in organoids. Immunofluorescence analysis revealed the familiar apical enrichment of CEA in the Colon-Chip (strong) and Duodenum-chip (weak) (*Figure 4—figure supplement 3B*), whereas CEA appeared to be junctional and non-polarized in colon organoids. Indeed, the incorporation of microenvironmental cues, including flow and peristalsis, has shown to lead to enhanced maturation within the Intestine-Chip (*Kasendra et al., 2020*; *Kasendra et al., 2018*). Importantly, immunohistochemistry analysis of a commercial 2D model comprising primary intestinal epithelial cells showed patchy CEA expression, which appeared weaker than that observed in the native colon (*Figure 4—figure supplement 3C*). Consistently, we estimated an average of about 2000 CEA surface binding sites per cell in this system (*Figure 4—figure supplement 3D*), which is dramatically lower than the expression recorded in the Colon Intestine-Chip.

Next, we evaluated the Intestine-Chip for its ability to capture the toxic effects CEA(Hi) and CEA (Lo) TCB, and expected differences therein, owing to differential binding affinity. We focused first on the Colon Intestine-Chip, bearing in mind the higher levels of target in the native organ and the chip model. To render the Intestine-Chips capable of simulating an immune response, we took an approach analogous to that of the Lung-Chip: PBMC and TCBs were introduced using the top fluidic channel of the Colon Intestine-Chip, affording direct contact with the epithelium and enabling engagement with the target (*Figure 4—figure supplement 2*). Epithelial cell death, immune cell attachment, and activation were monitored as readouts of on-target TCB safety liabilities. Unlike the Alveolus Lung-Chip, TCB treatment did not lead to increased epithelial cell killing. Nonetheless, we did observe significant changes in all of the other readouts of TCB-mediated toxicity. Both CEA(Hi) and CEA(Lo) TCB induced dose-dependent increase in PBMC attachment (*Figure 4D*) and activation, as evidenced by CD69 upregulation (*Figure 4E*) and the release of pro-inflammatory cytokines, including granzyme B, IFNγ, and IL-6 (*Figure 4F*). As expected, CEA(Hi) TCB triggered higher PBMC attachment, activation and cytokine release compared with CEA(Lo) TCB, confirming that the model is sensitive to differences in antibody affinity, which would be expected to translate into differential toxicity outcomes in the clinic. Importantly, we observed no activation upon treatment with an antibody that engages the CD3 receptor on T cells but cannot bind to CEA (NT TCB). Likewise, the CEA-targeting TCBs failed to induce activation of PBMC only, in the absence of target tissue (*Figure 4—figure supplement 3E*). Together, these data confirm that the effects observed in the chips

are target dependent and mode of action dependent, ruling out non-specific PBMC activation and cytokine release mediated by CD3 engagement only.

We then took advantage of the Duodenum Intestine-Chip, which faithfully mimics the lower CEA expression observed in the native small intestine (*Figure 4B,C*), to explore whether the system is sensitive to variations in target expression. Indeed, we observed differences in on-target toxicity governed by target abundance: CD69 expression and cytokine release induced by both TCBs were significantly attenuated in the Duodenum Intestine-Chip, in line with the lower target expression (*Figure 4G–H*). The reduction was more extensive in the case of CEA(Lo) TCB, which induced minimal increase in CD69 expression and no cytokine release, relative to the non-targeting TCB. CEA(Hi) TCB induced PBMC activation and cytokine release which, while lower than those observed in the Colon Intestine-Chip, were significantly elevated compared with the non-targeting control, suggesting that the high-affinity molecule may pose safety risks even in tissues with low target expression as described for the high-affinity FOLR1 molecule in the lung.

## Discussion

Here, we describe an Organs-on-Chips based approach for the assessment of TCB toxicity in the lung and the intestine. We demonstrate that the Alveolus Lung-Chip successfully recapitulated FOLR1 TCB-mediated lung toxicities observed in cynomolgus monkeys and instructed the design of a second-generation molecule, whose favorable lung safety profile predicted by the chip model was verified in vivo. The Intestine-Chip model captured the liabilities of a TCB targeting a human-specific antigen, thus filling the gap left by the lack of cross-reactivity in animals and suitable animal models overall. The model likewise displayed sensitivity to TCB affinity and predicted differential, target expression-dependent toxicity outcomes between different intestinal regions. Both models were able to shed light into the toxicity mechanisms of the molecules, clearly decoupling target-mediated effects from T cell activation through CD3 engagement only, which has been shown to be an additional mode of TCB-induced adverse events (*Segal et al., 1999*). It is worth mentioning that both the Lung-Chip and the Intestine-Chip demonstrated an advantage over conventional models for this application: the Alveolus Lung-Chip was found to report more specific TCB-responses and provide additional important readouts compared with transwell-based approaches, whereas the Intestine-Chip supported more physiologically relevant target expression, compared with both 3D organoids and primary intestinal barrier grown in transwells.

Owing to the absence of predictive early-stage assays, 'on-target, off-tumor' TCB safety liabilities have in some cases been only detected either in late-stage preclinical models (non-human primates) or as life-threatening adverse events in the clinic. Advanced human cellular models that capture the immunopathology of TCB-induced adverse events would aid the iterative antibody design at the early stage, thus ensuring favorable safety profiles before entering clinical trials, reducing attrition rates, and ultimately expediting the application of these potentially life-saving therapies. Importantly, the mechanistic insights into on-target, off-tumor toxicities and design opportunities afforded by these platforms are not restricted to T-cell engagers like TCBs, but also apply to chimeric antigen receptor T (CART) cell therapy, bearing in mind their similar modes of action. For instance, a human epidermal growth factor receptor 2 (Her2) CART therapy, intended to treat a patient with colorectal cancer, led to lethal toxicity through off-tumor cardiopulmonary targeting (*Morgan et al., 2010*). To improve its efficacy-safety profile, this therapy was affinity-tuned to detect tumor cells with a high density of surface antigens, while sparing normal cells with lower antigen expression (*Zhao et al., 2009*). A mouse model expressing human Her2 was used to confirm the safer profile of the low-affinity CART (*Castellarin et al., 2020*). Using the Lung- and Intestine-Chips, we similarly identified a reduced risk of healthy tissue targeting with a low-affinity TCB for both the FOLR1 and CEA targets. However, in contrast to the humanized mouse models, these in vitro tools are fully human, applicable to various targets and faster to generate, making them a promising alternative for antibody/CART preclinical safety testing, format selection, and optimization.

Furthermore, the immune-competent Organs-Chips described here can bridge preclinical research to clinical application, by aiding the discovery of early predictive biomarkers of TCB/CART toxicity in patients, which would certainly help to anticipate and manage life-threatening adverse events. While no universally predictive markers of adverse events associated with these therapies are currently accepted and validated, the phenotypic outcomes observed in the chips closely match

some of the few clinical indicators that have been proposed. For example, early elevation of specific serum cytokines, including IFN-γ and IL-6, was found to precede the development of severe cytokine release syndrome in response to CART therapy (*Teachey et al., 2016*; *Hay, 2018*). The release of IFN-γ and IL-6 was consistently observed in both the Alveolus Lung and Colon Intestine-Chip upon treatment with high-affinity FOLR1 and CEA TCB, which led to on-target toxicity. The detection of such clinically relevant biomarkers exemplifies the translational value of our platform. Going forward, the system could be coupled with, for example, unbiased proteomic analyses of chip supernatants and transcriptomic dissection of the effector and target cell pools and used to uncover early novel predictors or TCB-mediated adverse events. It may also be of interest to other types of molecules such as super agonist CD28 antibodies, which induced a severe cytokine storm in the clinics (*Hünig, 2012*).

Another attractive future development of our platform is its use to evaluate the therapeutic window of a therapy. We tested the same molecule in a healthy Lung-Chip versus a cancer-Chip and found a 1000-fold difference in the concentration of TCB required to kill target cells (*Figure 2—figure supplement 2*). As a next step, an all-in-one platform could combine a healthy population with a tumor one in the same chip as previously described (*Hassell et al., 2017*), thus capturing safety liabilities triggered by tumor lysis itself. Although efficacy–safety interactions are clinically relevant, these two aspects are often evaluated independently prior to phase I clinical studies, owing largely to the absence of fitting preclinical tools. We believe that using a chip comprising only healthy tissue is a reasonable safety evaluation approach in the case when the organ of concern is different from the cancer-affected organ. However, the tumor microenvironment and anti-tumor efficacy effects might influence the toxicity outcome when the target-expressing organ and the tumor co-localize, and should therefore be taken into account. For example, treating colorectal cancer with CEA-targeting TCBs may lead to an increased risk of intestinal toxicities, taking into consideration that tumor cell lysis may lead to the release of inflammatory cytokines and tumor antigens, thus further potentiating immune cell activation and cytotoxicity. The coupling of healthy and tumor biopsy-derived organoids with immunocompetent Organ-Chip technology may provide the ideal setup for a combined efficacy and safety assessment. Aside from safety questions, the presence of a relevant tumor compartment would open possibilities for tuning of compound pharmacology and efficacy testing. For example, a matrix layer below a tumor cell layer could be used to better recapitulate the immunosuppressive tumor environment via the introduction of regulatory T cells and tumor-associated macrophages or fibroblasts, taking it into account when designing therapies.

In conclusion, we describe novel human immunocompetent models of the lung and intestine and validate them as platforms for TCB safety profiling, outlining how these systems could reduce the reliance on animal-based safety assessments, enable educated antibody format selection, shed light into the mechanistic underpinnings of toxicities, and support the identification of clinical biomarkers. Going forward, the concepts we introduced here can be expanded to address the persisting gap in modeling immune-related toxicities, associated with, for example, immune checkpoint blockade (*Brahmer et al., 2018*; *Ramos-Casals et al., 2020*). Considering their systemic and multifactorial immunopathology, modeling these processes accurately would likely require the incorporation of resident immune cells and lymphoid structures, as well as modalities that support the simulation of T-cell trafficking and tissue infiltration.

## Materials and methods

**Key resources table**

| Reagent type (species) or resource | Designation | Source or reference | Identifiers | Additional information |
|---|---|---|---|---|
| Biological sample (*Homo sapiens*) | Human Primary Alveolar Epithelial Cells | CellBiologics | Cat# H-6053 | |
| Biological sample (*Homo sapiens*) | Human Pulmonary Alveolar Epithelial Cells | Accegen | Cat# ABC-TC3770 | |

*Continued on next page*

*Continued*

| Reagent type (species) or resource | Designation | Source or reference | Identifiers | Additional information |
|---|---|---|---|---|
| Biological sample (*Homo sapiens*) | Human Lung Microvascular Endothelial Cells | Lonza | Cat# CC-2527 | |
| Biological sample (*Homo sapiens*) | Human Large Intestine Microvascular Endothelial Cells | Cell Systems | Cat# ACBRI 666 | |
| Cell line (*Homo sapiens*) | HeLa | ATCC | Cat# CCL-2, RRID:CVCL_0030 | |
| Cell line (*Homo sapiens*) | MKN45 | DSMZ | Cat# ACC-409, RRID:CVCL_0434 | |
| Cell line (*Homo sapiens*) | HEK293T | ATCC | Cat# CRL-11268, RRID:CVCL_1926 | |
| Commercial assay or kit | QIFIKIT | Agilent | Cat# K007811-8 | |
| Commercial assay or kit | Cell Tracker green | ThermoFisher | Cat# C7025 | |
| Commercial assay or kit | NucView405 Caspase-3 Enzyme | Biotium | Cat# 10407 | |
| Commercial assay or kit | ProcartaPlex multiplex immunoassays | Invitrogen | Cat# PPX-12-MXNKRV6 | |
| Software, algorithm | Prism | GraphPad | | |
| Software, algorithm | Fiji | | RRID:SCR_002285 | |

## Construction of FOLR1- and CEA-targeted molecules

FOLR1 and CEA-targeted TCB molecules were generated in the 2+1 format (two target binding Fabs and one CD3 binding Fab). Heterodimerization of these bispecific antibodies is achieved by using the 'knob-into-hole' technology (*Ridgway et al., 1996*) in which the FOLR1 Fab is N-terminally fused to the CD3-Fc knob chain (head-to-tail configuration) and a second FOLR1 Fab is fused to the Fc hole chain. These antibodies lack Fc effector functions due to the insertion of the PG LALA mutations (P329G; L234A, L235A; *Schlothauer et al., 2016*). In the FOLR1 TCB molecule, a common light chain was used for both, the FOLR1 and CD3 Fab. Two versions of the FOLR1 TCB were generated which differ in affinity. Clone 16D5 reveals high affinity (nM), whereas a variant of this clone, carrying two amino acid changes (D52dE and W96Y according to Kabat numbering), exhibits an affinity in the µM range.

The CEA TCB were generated analogously but use different light chains for CEA and CD3 Fabs. To avoid light chain mispairing, the CrossMabVH-VL technology (*Klein et al., 2012*) was applied generating a VH/VL crossover in the CD3 Fab and a corresponding crossed light chain (VHCL). In addition, charged residues were introduced in the constant kappa and CH1 domains of the CEA Fabs to furthermore force correct light chain pairing. Two versions of the CEA-targeted TCB were generated which differ in affinity.

The genes for all chains of the TCB molecules were inserted into separate mammalian expression cassettes by standard recombinant DNA technologies and expressed either transiently in HEK293 cells or in stable CHO clones. Purification of bispecific TCB molecules was performed according to standard Protein A affinity and size exclusion methods.

For both molecules, non-tumor targeted TCBs were generated based on the same construct but without the FOLR1 or CEA targeting sequence. Their characterization was done as described previously in Jurkat NFAT reporter assay (for quantification of CD3 downstream signaling) and in various cell lines cytotoxic assays (for non-tumor targeting assessment) (*Geiger et al., 2020*; *Seckinger et al., 2017*).

## Cell lines

The cell lines used were as follows: HeLa (ATCC CCL-2), MKN45 (DSMZ ACC 409), HEK293T (ATCC CRL-11268). Cell lines were maintained by the supplier. No additional authentication was performed by the authors of this study. Cell lines were tested for mycoplasma contamination and shown to be free from mycoplasma. No commonly misidentified cell lines were used.

## Patient-derived xenograft (PDX) model

All mice were maintained under specific pathogen-free condition with daily cycles of 12 hr light/12 hr darkness. The animal facility has been accredited by the Association for Assessment and Accreditation of Laboratory Animal Care (AAALAC). All animal studies were performed in accordance with the Federation for Laboratory Animal Science Associations (FELASA). The animal studies were approved by and done under license from the Government of Upper Bavaria (Regierung von Oberbayern; Approval number: Az 55.2.1.54–2532.0-10-16). We have complied with all relevant ethical guidelines and regulations. Animals were maintained for 1 week after arrival to get accustomed to the new environment and for observation. Daily continuous health monitoring and weekly body weight measurement was conducted.

Female NSG mice were injected intraperitoneal with 15 mg/kg of Busulfan followed one day later by an intravenous injection of $1 \times 10^5$ human hematopoietic stem cells isolated from cord blood. At weeks 14–16 after stem cell injection, mice were bled sublingual and blood was analyzed by flow cytometry for successful humanization. Efficiently engrafted mice were randomized according to their human T-cell frequencies into the different treatment groups. The human breast cancer patient-derived HER2+ ER− xenograft model BC_004 was purchased from OncoTest (Freiburg, Germany). Tumor fragments were digested with Collagenase D and DNase I (Roche), counted and $2\times10^6$ BC004 cells were injected in total volume of 20 μL PBS into the mammary fat pad of humanized NSG mice. Treatment with FOLR1 TCB started once weekly at a dose of 0.5 mg/kg when tumor size reached approximately 400 mm³ (day 28). Control group received histidine buffer (Vehicle). All mice were injected intravenously with 200 μL of the appropriate solution.

Alternatively for CEA TCB assessment, mice were injected with $1 \times 10^6$ MKN45 cells in the subcutaneous right flank and treated once weekly (CEA(Hi) TCB) or twice weekly (CEA(Lo) TCB) at a dose of 0.5 or 2.5 mg/kg respectively when tumor size reached approximately 150 mm³ (day 7). Control group received a histidine buffer (Vehicle). All mice were injected intravenously with 200 μL of the appropriate solution.

## Alveolus Lung-Chip

### Immunohistochemistry cynomolgus monkey and human tissues

Immunohistochemical staining for FOLR1 distribution in cynomolgus monkey or human formalin-fixed, paraffin-embedded tissues was carried out on a Discovery XT automated slide stainer using a mouse anti-human monoclonal antibody for FOLR1 (Novocastra Clone BN3.2; Leica Biosystems, Wetzlar, Germany) at 15 μg/mL after antigen retrieval with Cell Conditioning 1 (CC1; Ventana Medical Systems Inc). As secondary Antibody was used a donkey anti-mouse biotinylated polyclonal IgG (Jackson Immunoresearch Lab, cat: 715-065-151) at 6 μg/mL. DAB Map Kit (Ventana 760–124) was used as a detection system. Xenograft tumors from FOLR1-expressing HeLa cells were used as a positive control.

## Cell culture

Human alveolar epithelial cells (Cell Biologics, Accegen) were cultured using SABM medium (Lonza) supplemented with growth factor kit and 5% v/v fetal bovine serum (FBS) in a T-25 flask coated with Gelatin (ATCC) until they reach 90% confluency.

Human microvascular lung endothelial cells (HMVEC-L) (Lonza) were cultured in EBM-2 Basal Medium supplemented with EGM-2 MV Microvascular Endothelial Cell Growth Medium and 1% v/v Pen-Strep (ThermoFisher) according to manufacturer's instructions.

Peripheral blood mononuclear cells (PBMC) were isolated from fresh human buffy coat (Research Blood Components) using immunomagnetic negative selection (Stem Cell Technologies) and cultured in RPMI-1640 (Gibco) supplemented with 10% v/v FBS (ThermoFisher) and 1% v/v Pen-Strep (ThermoFisher) or cryo-preserved in FBS containing 10% dimethyl sulfoxide (DMSO) before use.

## Alveolus Lung-Chip

The design and fabrication of Organ-Chips has been previously described (*Huh et al., 2012*). Briefly, the S-1 Chips are composed of transparent polydimethylsiloxane (PDMS) containing two parallel microchannels: an epithelial channel (1 × 1 mm) and vascular channel (200 μm × 1 mm) separated by a porous membrane. The chip protocol was performed according to the manufacturer's instructions (Alveolus Lung-Chip Culture Protocol, Emulate Inc). S-1 chip microchannels were functionalized to covalently attach extracellular matrix proteins (ECM) before seeding using ER solutions (Emulate Inc). Chip channels were then coated with a mixture of ECM in Dulbecco's phosphate-buffered saline (DPBS): 200 μg/mL human placenta collagen type IV (Sigma-Aldrich) and 30 μg/mL fibronectin (Gibco) for the vascular channel, and 200 μg/mL human placenta collagen type IV (Sigma-Aldrich) and 30 μg/mL human plasma fibronectin (Corning) and 5 μg/mL human placenta laminin (Sigma-Aldrich) for the epithelial channel. Chips were then incubated overnight at 37°C for coating and channels were washed next day with their respective growth medium before seeding. Human Alveolar Epithelial Cells (HPAECs) were seeded at a density of $0.5 \times 10^6$ cells/mL following protocols (Alveolus Lung-Chip Culture Protocol, Emulate Inc). Human Microvascular Lung Endothelial Cells (HMVEC-L) were seeded at a density of $5 \times 10^6$ cells/mL following the Alveolus Lung-Chip protocol (Emulate Inc). Air-Liquid Interface is introduced on day 5 of culture following the protocol and maintained for 4 days. On the day before dosing, hydrocortisone was removed from the bottom channel growth medium.

## PBMC administration

After 4 days of culture under air-liquid interface, culture medium was re-introduced in the epithelial channel before dosing with PBMC-TCB. 500 μL of dosing media (M199 +2% v/v FBS) was added to the epithelial inlet reservoir. Liquid–Liquid interface was re-introduced at 1000 μL/hr for 5 min on the epithelial channel, keeping the vascular channel at 0 after which the flow was switched to 30 μL/hr in both channels.

Frozen PBMC after thawing were suspended overnight at $4 \times 10^6$ cells/mL in complete RPMI-1640 medium with 10% v/v FBS. The viability of PBMC was determined by using trypan blue exclusion assay. PBMC were allowed to rest overnight at 37°C. The following day, PBMC were stained using cell tracker green (ThermoFisher) according to the manufacturer's instructions. PBMC dosing solutions (Dosing media: M199 (ThermoFisher) + 2% v/v FBS) were prepared by incubating cell suspensions at $2 \times 10^6$ cells/mL in media containing TCBs at different concentrations for an hour at 37°C prior to administration. After the incubation period, the epithelial channel inlets were aspirated and 500 μL of dosing solution was added to the inlet. PBMC were administered to the chips at 1000 μL/hr for 5 min. After PBMC administration, the system was left static for 3 hr before starting flow at 30 μL/hr with fresh dosing media (M199 + 2% v/v FBS) without TCBs in the epithelial channel and custom ALI media without hydrocortisone (Alveolus Lung-Chip Protocol, Emulate Inc) in the vascular channel.

## Transwells

Twenty-four-well sterile transwell (Corning) inserts with polyester membrane 0.4 μM pore size was used. Similar to the chips, transwell inserts were coated with a mixture of ECM in Dulbecco's phosphate-buffered saline (DPBS): 200 μg/mL human placenta collagen type IV (Sigma-Aldrich) and 30 μg/mL fibronectin (Gibco) for the vascular or bottom compartment, and 200 μg/mL human placenta collagen type IV (Sigma-Aldrich) and 30 μg/mL human plasma fibronectin (Corning), and 5 μg/mL human placenta laminin (Sigma-Aldrich) for the epithelial or top compartment. Transwells were then incubated overnight at 37°C for coating and channels were washed next day with their respective growth medium before seeding. Human alveolar epithelial cells (HPAECs) were seeded at a density of $0.1 \times 10^6$ cells/well following protocols (Alveolus Lung-Chip Culture Protocol, Emulate Inc). Human microvascular lung endothelial cells (HMVEC-L) were seeded at a density of $0.1 \times 10^6$ cells/well following the Alveolus Lung-Chip protocol (Emulate Inc). Air–liquid interface is introduced on day 5 of culture following the same protocol as the Alveolus Lung-Chip. After 4 days of culture under air–liquid interface, epithelial cells were dosed with PBMC-TCB. 100 μL of PBMCs-TCB mixture stained with cell tracker green were added to the epithelial compartment at $2 \times 10^6$ cells/mL and

500 µL of custom ALI media without hydrocortisone (Alveolus Lung-Chip Protocol, Emulate Inc) in the vascular compartment.

For timepoints, T = 24 and 48 hr after PBMC administration, 50 µL of supernatant was collected from the epithelial and vascular compartments for further cytokine analysis. NucView405 Caspase-3 Enzyme, fluorescent caspase 3/7 substrate for detecting apoptosis by live staining (Biotium) at 2 µM was prepared with dosing medium (M199 +2% v/v FBS). 100 µL of the live stain was added to the epithelial compartment and incubated in 37°C for 30 min. After which the epithelial compartment was carefully aspirated leaving some media and replaced with fresh medium. Transwells were then transferred to fluorescent microscope (Olympus IX83 Inverted Microscope) for live imaging. Additional brightfield images were captured using the Echo Revolve microscope.

PBMC were harvested from the Alveolus Lung Transwells epithelial compartment at T=48 hr (terminal timepoint) after dosing with PBMC-TCB, by repeated washing using 200 µL tips. The PBMC suspension was then transferred to a V bottom 96-well plate. The staining protocol followed for flow cytometry analysis was the same as in the Alveolus Lung-Chips. Sample data was acquired using BD FACSCelestaTM flow cytometer (BD BioSciences), and data was analyzed using FlowJo V10 software (FlowJo).

## Target expression

For quantification of target expression, HPAECs (day 0, day 5, day 10 of chip culture) were recovered using TrypLE Express Enzyme (Gibco) at 37°C for 10 min. Epithelium from Alveolus Lung-Chips cultured to day 5 and day 10 was obtained by filling both channels with TrypLE solution incubating at 37°C until complete dissociation was achieved using gentle pipetting. The dissociated epithelium was collected from the epithelial channel and digestion was quenched using SAGM culture medium with 2% v/v FBS. All single-cell samples were distributed at $0.5 \times 10^6$ cells/mL for live staining with monoclonal mouse anti-human FOLR1 IgG1 (LS Bio) in DPBS with 2% v/v FBS (Sigma). Secondary staining for target was performed using QIFIKIT (BIOCYTEX) anti-mouse IgG, along with mouse IgG1 Isotype FOLR1 (L.S Bio) for secondary control and provided calibration and standard beads. Samples were run with BD FACSCelesta flow cytometer (BD Biosciences), and data analyzed using FlowJo V10 software (FlowJo).

## RNA isolation

Total RNA was isolated from the Alveolus Lung-Chip using TRIzol reagent (Sigma) following manufacturer's instructions and flash frozen in liquid nitrogen. Samples were sent to GENEWIZ for sequencing.

## RNA sequencing bioinformatics

The RNA sequencing was performed using the Illumina TruSeq paired-end sequencing platform with read length 2× 150 bp and sequencing depth ~28M paired end reads/sample. To remove poor quality adapter sequences and nucleotides, we trimmed the sequence reads using the Trimmomatic v.0.36. The STAR (Spliced Transcripts Alignment to a Reference) aligner v.2.5.2b was used to map the trimmed reads to the *Homo sapiens* reference genome GRCh38 (available on ENSEMBL) and generate the BAM files. Using the featureCounts from the Subread package v.1.5.2 we calculated the unique gene hit counts. Only unique reads that fell within exon regions were counted. Note that since a strand-specific library preparation was performed, the reads were strand-specifically counted. Using the gene hit counts and the corresponding gene lengths we calculated the FPKM (Fragments Per Kilobase of exon per Million reads mapped) gene expression levels.

## Live staining and imaging

For timepoints T = 24 and 48 hr after PBMC-TCB administration, effluents were collected for further analysis and pod inlets were aspirated. NucView405 Caspase-3 Enzyme, fluorescent caspase 3/7 substrate for detecting apoptosis by live staining (Biotium) at 2 µM was prepared with dosing medium (M199 +2% v/v FBS). 500 µL of the live stain was added to the epithelial channel inlet reservoirs. Epithelial channel of the chips was flowed at 1000 µL/hr for 5 min while setting the vascular channel to 0. Flow was then reset to 30 µL/hr for both the channels for 30 min, fresh media was then flushed through after incubation. Chips were then transferred to fluorescent microscope (Olympus IX83

Inverted Microscope) one at a time for live imaging. Additional brightfield images were captured using the Echo Revolve microscope.

## Flow cytometry

PBMC were harvested from the Alveolus Lung-Chip epithelial channel at T=48 hr (terminal timepoint) after dosing with PBMC-TCB, by repeated washing using 200 μL tips by blocking the chip inlet. PBMC from each chip was transferred to a V bottom 96-well plate and washed with DPBS + 1% v/v FBS solution before staining with surface markers. Master mix of surface markers was prepared in Brilliant Buffer solution (BD BioSciences) which consisted of anti-human Alexa Fluor 700 anti-human CD3 (BioLegend, cat. 300324), Brilliant Violet 785 anti-human CD4 (BioLegend, cat. 317442) and Brilliant Violet 650 anti-human CD69 (BioLegend, cat. 310934). Harvested PBMC was stained with the prepared master mix for 20 min at 4°C and fixed using 1% v/v paraformaldehyde in DPBS for 15 min at room temperature. Samples collected were then washed with DPBS + 1% v/v FBS solution and stored in 4°C and read within 3 days.

Sample data was acquired using BD FACSCelestaTM flow cytometer (BD BioSciences) and data was analyzed using FlowJo V10 software (FlowJo).

## Immunofluorescence microscopy

For alveolar epithelial cell staining, samples were fixed with 4% paraformaldehyde (Electron Microscopy Sciences) for 20 min at room temperature. Samples were then washed twice with DPBS and perfused with 100 mM glycine to quench autofluorescence for 30 min at room temperature, then rinsed with DPBS and permeabilized with 0.1% v/v Triton-X for 10 min and blocked with 1% v/v BSA and 5% v/v Normal Donkey serum in DPBS for 30 min. Samples were then stained with primary antibodies overnight at 4°C, with the following primary antibodies diluted 1:100 in 2% v/v BSA in DPBS and then rinsed twice with DPBS before staining with secondary antibodies diluted 1:200 in 2% v/v BSA in DPBS for 2 hr in the dark at room temperature, and counterstained with NucBlu (Thermo-Fisher) following the manufacturer's instructions. The primary antibodies used were rabbit polyclonal anti-E-Cadherin (abcam), mouse monoclonal anti-human FOLR1 IgG1 (LS Bio). Secondary antibodies used were donkey anti mouse or rabbit Alexa Fluor 488, Alexa Fluor 568, Alexa Fluor 647-conjugated antibodies (Abcam), goat anti-mouse IgM (Heavy chain) Alexa Fluor 488 (ThermoFisher, A-21042). Immunofluorescence microscopy was performed using an Inverted Olympus IX83 microscope and Echo Revolve (Echo). At least 5 fields of view were taken per chip along the co-culture channel.

## Image analysis

Image analysis was performed using ICY software (BioImage Analysis Lab, Institut Pasteur) to quantify PBMC attachment (CellTracker Green) to the alveolus epithelium and apoptotic (NucView405 Caspase-3) alveolar epithelial cells.

Using co-localization tools, the number of apoptotic epithelial-positive cells (Caspase-3$^+$ NucView405$^+$) that are also PBMC cell tracker negative (GFP$^-$) was quantified and setting different object sizes for alveolar epithelial cells (~900–3000 pixels) and PBMCs (~200–600 pixels) in the ICY image analysis software. This image analysis tool could give us a better estimate of epithelium death by apoptosis under PBMC exposure to the Alveolus Lung-Chip.

## Cytokine analysis

At T = 24 and 48 hr after PBMC-TCB administration, effluents were collected from Alveolus Lung-Chip Pod outlets. Effluents were then immediately frozen at −80°C until measurement. Measurement of cytokines for Alveolus Lung-Chip (GranzymeB, IFNγ, IL-2, IL-6, IL-8, IL-10, IL-13, IL1RA, TNFα, MIP-1β, G-CSF, GM-CSF) was performed using customized Invitrogen ProcartaPlex multiplex immunoassays (reference PPX-12-MXNKRV6). Each kit contained a black 96-well plate (flat bottom plate), antibody-coated beads, detection antibody, streptavidin-R-phycoerythrin (SAPE), reading buffer and universal assay buffer. In addition, standards with known concentration were provided to prepare a standard curve. According to the Invitrogen Publication Number MAN0017081 (Revision B.0 (33)), the assay workflow was the following. After adding the beads into the flat bottom plate, the beads were washed using a flat magnet and an automated plate washer (405TS microplate washer from

Bioteck). Then standards and samples diluted with a universal buffer were added into the plate and a first incubation started for 2 hr. After a second wash, detection antibodies were added. After 30 min incubation and a wash, SAPE was added. Finally, after 30 min incubation and a last wash, the beads were resuspended in the reading buffer and the plates were ready for analysis.

The data was acquired with a Luminex instrument, BioPlex-200 system from Bio-Rad. Using the Certificate of Analysis provided with the kit, bead region and standard concentration value S1 for each analyte of the current lot were entered in the software, BioPlex Manager. Plotting the expected concentration of the standards against the mean fluorescent intensity (MFI) generated by each standard, the software generated the best curve fit and calculated the concentrations of the unknown samples (in pg/mL). The data were then exported in Excel and plotted in Graphpad Prism.

## Surface plasmon resonance

The avidity of the interaction between the anti-FOLR1 T-cell bispecifics and the recombinant folate receptors was determined as described below. Recombinant biotinylated monomeric Fc fusions of human and cynomolgus Folate Receptor 1 (FOLR1-Fc, produced in house) were directly coupled on a SA chip using the standard coupling instruction (Biacore, Cytiva). The immobilization level was about 200–300 RU. The anti-FOLR1 T-cell bispecifics were passed at a concentration range from 11.1 to 900 nM with a flow of 30 µL/min through the flow cells over 180 s. The dissociation was monitored for 240 or 600 s. The chip surface was regenerated after every cycle using a double injection of 30 s 10 mM glycine–HCl pH 1.5. Bulk refractive index differences were corrected for by subtracting the response obtained on reference flow cell immobilized with recombinant biotinylated murine IL2R Fc fusion (unrelated Fc fused receptor). The binding curves resulting from the bivalent binding of the T-cell bispecifics were approximated to a 1:1 Langmuir binding (even though it is a 1:2 binding) and fitted with that model to get an apparent KD representing the avidity of the bivalent binding. The apparent avidity constants for the interactions were derived from the rate constants of the fitting using the Bia Evaluation software (Cytiva).

The affinity of the interaction between the anti-FOLR1 T-cell bispecifics and the recombinant folate receptors was determined as described below. For affinity measurement, direct coupling of around 12,000 resonance units (RU) of the anti-human Fab specific antibody (Fab capture kit, Cytiva) was performed on a CM5 chip at pH 5.0 using the standard amine coupling kit (Cytiva). Anti-FOLR1 T-cell bispecifics were captured at 20 nM with a flow rate of 10 µL/min for 40 s, the reference flow cell was left without capture. Dilution series (12.3–3000 nM) of human and cyno Folate Receptor 1 Fc fusion were passed on all flow cells at 30 µL/min for 240 s to record the association phase. The dissociation phase was monitored for 300 s and triggered by switching from the sample solution to HBS-EP. The chip surface was regenerated after every cycle using a double injection of 60 s 10 mM glycine–HCl pH 2.1. Bulk refractive index differences were corrected for by subtracting the response obtained on the reference flow cell 1. The affinity constants for the interactions were derived from the rate constants by fitting to a 1:1 Langmuir binding using the Bia Evaluation software (Cytiva).

## Binding of FOLR1-targeted TCBs to human FOLR1-expressing tumor cells

Experiments were performed with n=4 chips per condition and 8 to 10 fields of view per chip. All graphs are plotted as group means (individual points displayed if n<5 samples per group) ± SEM. Statistical significance (p<0.05) was determined by one-way or two-way ANOVA using Tukey's multiple comparison test.

## Assessment of FOLR1 TCBs binding to human FOLR1 expressed on HeLa cells

The binding of FOLR1 TCBs to human FOLR1 was assessed on HeLa cells. Briefly, cells were harvested, counted, checked for viability and resuspended at $2 \times 10^6$ cells/mL in FACS buffer (100 µL PBS 0.1% BSA). 100 µL of cell suspension (containing $0.2 \times 10^6$ cells) was incubated in round-bottom 96-well plates for 30 min at 4°C with different concentrations of the bispecific antibodies (30 pM–500 nM). After two washing steps with cold PBS 0.1% BSA, samples were re-incubated for further 30 min at 4°C with FITC-conjugated AffiniPure F(ab')2 Fragment goat anti-human IgG Fcg Fragment Specific

secondary antibody (Jackson Immuno Research Lab PE # 109-096-098). After washing the samples twice with cold PBS, samples were resuspended in PBS 0.1% BSA and analyzed on a FACS Canto II (Software FACS Diva). Binding curves were obtained using GraphPadPrism6.

## TCB-mediated lysis of tumor cells in vitro

T-cell killing mediated by FOLR1 TCBs was assessed on HeLa (high FOLR1) cells. Human PBMC were used as effectors and the killing was detected at 24 hr of incubation with the bispecific antibodies. Briefly, target cells were harvested with Trypsin/EDTA, washed, and plated at a density of 25,000 cells/well using flat-bottom 96-well plates. Cells were left to adhere overnight. Peripheral blood mononuclear cells (PBMC) were prepared by Histopaque density centrifugation of enriched lympho-cyte preparations (buffy coats) obtained from healthy human donors. Fresh blood was diluted with sterile PBS and layered over Histopaque gradient (Sigma, #H8889). After centrifugation (450 × g, 30 min, room temperature), the plasma above the PBMC-containing interphase was discarded and PBMC transferred in a new falcon tube subsequently filled with 50 mL of PBS. The mixture was cen-trifuged (400 × g, 10 min, room temperature), the supernatant discarded and the PBMC pellet washed twice with sterile PBS (centrifugation steps 350 × g, 10 min). The resulting PBMC population was counted (ViCell) and stored in RPMI1640 medium containing 10% FCS and 1% L-alanyl-L-gluta-mine (Biochrom, K0302) at 37˚C, 5% $CO_2$ in cell incubator until further use. For the killing assay, the antibody was added at the indicated concentrations (range of 0.01 pM–10 nM in triplicates). PBMC were added to target cells at final E:T ratio of 10:1. Target cell killing was assessed after 24 hr of incubation at 37˚C, 5% $CO_2$ by quantification of LDH released into cell supernatants by apoptotic/necrotic cells (LDH detection kit, Roche Applied Science, #11644793001). Maximal lysis of the target cells ( = 100%) was achieved by incubation of target cells with 1% Triton X-100. Minimal lysis ( = 0%) refers to target cells co-incubated with effector cells without bispecific construct.

## Statistics

Experiments were performed with n=4 chips per condition and 8–10 fields of view per chip. All graphs are plotted as group means (individual points displayed if n<5 samples per group) ± SEM. Statistical significance (p<0.05) was determined by one-way or two-way ANOVA using Tukey's multi-ple comparison test unless specified otherwise.

## Intestine-Chip

### Binding of CEA-targeted TCBs to human CEA-expressing tumor cells

MKN45 (DSMZ ACC 409) cells were harvested using Cell Dissociation Buffer, washed once with PBS, and resuspended in FACS buffer (PBS + 0.1% BSA). 200,000 cells were seeded into a 96-well round bottom plate, the assay plate was centrifuged at 400 × g for 4 min, and the supernatant was removed. Antibody dilutions were prepared in FACS-buffer to cover a final concentration range of 0.03 nM – 500 nM (1:4 dilution steps). Cells were incubated with CEA(Hi) TCB and CEA(Lo) TCB for 30 min at 4˚C. FACS plates were washed twice with 150 µL FACS buffer and incubated with 25 µL of the FITC-labeled AffiniPure F(ab')2 Fragment Goat Anti-Human IgG secondary antibody (Jackson Immuno Research, 109-096-008; pre-diluted 1:40 with FACS buffer) for another 30 min at 4˚C. After two washing steps with FACS buffer, cells were fixed in FACS buffer, containing 2% paraformalde-hyde for 30 min at 4˚C. Finally, fluorescence was measured using BD FACS Canto II. EC50 values were calculated using GraphPadPrism.

## TCB-mediated lysis of tumor cells in vitro

TCB-induced lysis of CEA-positive target cells was assessed using MKN45 (DSMZ ACC 409) cells. Human PBMCs were used as effectors and the killing was detected at 24 hr and 48 hr of incubation with the bispecific antibodies. Briefly, target cells were harvested with Trypsin/EDTA, washed, and plated at a density of 30,000 cells/well using flat-bottom 96-well plates. Cells were left to adhere overnight. For the killing assay, the antibody was added at the indicated concentrations (range of 6 pM–100 nM for CEA(Lo) TCB and 1.3 pM–20 nM for CEA(Hi) TCB in triplicates). PBMCs were added to target cells at final ratio to tumor cells of 10:1. Target cell killing was assessed after 48 hr of incu-bation at 37˚C, 5% $CO_2$ by quantification of LDH released into cell supernatants by apoptotic/necrotic cells (LDH detection kit, Roche Applied Science, #11 644 793 001). Maximal lysis of the

target cells ( = 100%) was achieved by incubation of target cells with 1% Triton X-100. Minimal lysis ( = 0%) refers to target cells co-incubated with effector cells without bispecific construct.

## Quantification of T-cell activation in response to TCB treatment

Tumor cell lysis assay plates were centrifuged (400 × g for 4 min), cells were resuspended, washed with FACS buffer, and incubated with 25 µL of the diluted CD4/CD8/CD69 antibody mix for 30 min at 4°C (e.g. PE/Cy7 anti-human CD4 #557852, FITC anti-human CD8 #555634, APC anti-human CD25 #555434, as indicated). Cells were washed twice to remove unbound antibody and finally resuspended in 200 µL FACS buffer containing PI (propidium iodide) to exclude dead cells for the FACS measurement. Fluorescence was measured using BD FACS CantoII.

## Assessment of TCB binding to human- or cynomolgus monkey-derived CEA

HEK293T cells were transiently transfected to overexpress either human or cynomolgus monkey CEACAM5 were harvested using Cell Dissociation Buffer, washed once with PBS and resuspended in FACS buffer (PBS + 0.1% BSA). 100,000 cells were seeded into a 96-well round bottom plate, the assay plate was centrifuged at 400xg for 4 min, and the supernatant was removed. Antibody dilutions were prepared in FACS-buffer to cover a final concentration range of 7.6 pM–500 nM (1:4 dilution steps), respective 125 nM and 500 nM of the positive control antibody binding to cynomolgus monkey CEACAM5 (clone 28A9, internal production, ID AB03195). Cells were incubated with CEA (Lo) TCB or positive reference molecule for 30 min at 4°C. FACS plates were washed twice with 150 µL FACS buffer and incubated with 25 µL of the FITC-labeled AffiniPure F(ab')2 Fragment Goat Anti-Human IgG secondary antibody (Jackson Immuno Research, 109-096-008; pre-diluted 1:40 with FACS buffer) for another 30 min at 4°C. After two washing steps with FACS buffer, cells were stained with a live/dead dye (DAPI, diluted in PBS) for 30 min at 4°C. After a final washing step with FACS buffer, fluorescence was measured using a BD FACS CantoII.

## Cell culture

Human colon organoid cultures (colonoids) were established from biopsies obtained during surgical procedures utilizing methods developed by the laboratory of Dr. Hans Clevers (*Sato et al., 2011*). De-identified biopsy tissue was obtained from healthy adult subjects who provided informed consent at Johns Hopkins University, and all methods were carried out in accordance with approved guidelines and regulations. All experimental protocols were approved by the Johns Hopkins University Institutional Review Board (IRB). Routine expansion of colonoids was performed by embedding isolated intestinal crypts in droplets of growth factor–reduced Matrigel (Corning) and cultured in Human IntestiCult Organoid Growth Medium (StemCell) supplemented with 10 mmol/L Y-27632 (Sigma), 5 mmol/L CHIR99021 (ReproCell), and 50 mg/mL primocin (InvivoGen). After 3 days, colonoids Y-27632 and CHIR99021 supplements were removed. Colonoids were passaged every 7 days.

Human Large Intestine Microvascular Endothelial Cells (cHIMEC) (Cell Systems) were thawed at passage five and cultured in Endothelial Cell Growth Medium (EGM-2MV) (PromoCell) supplemented with Endothelial Cell Growth Medium MV2 Supplement Pack (PromoCell) and 1% v/v primocin (InvivoGen).

Peripheral blood mononuclear cells (PBMC) were isolated from fresh human buffy coats using immunomagnetic negative selection (Stem Cell Technologies) and cultured in RPMI-1640 (Gibco) supplemented with 10% v/v FBS and 1% v/v Pen-Strep or cryo-preserved in FBS containing 10% dimethyl sulfoxide (DMSO) before use.

## Colon Intestine-Chip culture

The design and fabrication of Organ-Chips has been previously described (*Huh et al., 2012*). Briefly, the S-1 Chips are composed of transparent polydimethylsiloxane (PDMS) containing two parallel microchannels: an epithelial channel (1 × 1 mm) and vascular channel (200 µm × 1 mm) separated by a porous membrane. S-1 chip microchannels were functionalized to covalently attach extracellular matrix proteins (ECM) before seeding using ER solutions (Emulate Inc) following provided protocols (Basic Organ-Chip Culture Protocol, Emulate Inc). Chip channels were then coated with a mixture of ECM in Dulbecco's phosphate-buffered saline (DPBS): 200 µg/mL human placenta collagen type IV

(Sigma-Aldrich) and 30 µg/mL fibronectin (Gibco) for the vascular channel; and 200 µg/mL human placenta collagen type IV (Sigma-Aldrich) and 100 µg/mL Matrigel (Corning) for the epithelial channel. Chips were then incubated overnight at 37°C for coating and channels were washed the next day with their respective growth medium. Human colonic microvascular endothelial cells (cHIMECs) were then seeded into the vascular channel at a density of $8 \times 10^6$ million cells/mL. After 1.5 hr, the chips were inverted and cHIMECs at a density of $8 \times 10^6$ million cells/mL were seeded again creating a contiguous vascular tube. Colonoids were recovered from Matrigel and fragmented as reported previously (*Kasendra et al., 2018*). Fragmented colonoids were suspended in organoid expansion medium at a density of 2–3 culture wells per chip and seeded onto the membrane of the epithelial channel.

The following day, vascular and epithelial channels were washed with EGM-2MV and organoid expansion medium, respectively, and connected in Pod portable modules (Basic Research Kit; Emulate, Inc). The Human Emulation System (Emulate Inc) was continuously perfused at 30 µL/hr for both channels with 2% cyclic stretching (0.15 Hz) from days 2 to 5, then with 10% cyclic stretching (0.15 Hz) until day 8 of culture. Supplements were removed from the epithelial channel media after day 2 of culture.

## PBMC administration

Twenty-four hours prior to PBMC-TCB administration, freshly isolated or thawed PBMC were suspended at $4 \times 10^6$ cells/mL in complete RPMI-1640 medium. The viability of PBMC was determined by using trypan blue exclusion assay. The acceptance criteria for PBMC viability was >85% to proceed to the next experimental step. PBMC were allowed to rest overnight at 37°C. The following day, PBMC dosing solutions were prepared by incubating cell suspensions in media containing TCBs at different concentrations for 4 hr at 37°C prior to administration. Epithelial channel TCB dosing media was also prepared by adding TCBs to organoid growth media. After the incubation period, the Pod epithelial channel inlets were aspirated and 500 µL of dosing solution (approx. $2 \times 10^6$ PBMC cells) was added to the inlet. PBMC were administered to the chips at 1000 µL/hr for 10 min. After PBMC administration, dosing media with and without TCBs was then added to the epithelial channel inlet and perfused through the chip. The vascular channels were perfused with EGM2-MV complete growth media.

## Target expression

For quantification of target expression, colonic and duodenum organoids (day 0 of chip culture) were recovered from Matrigel following standard procedure, then digesting in TrypLE Express Enzyme (Gibco) in DPBS at 37°C for 10 min to single cells. Epithelium from Colon and Duodenum Intestine-Chips cultured to day 5 and day 8 from the same organoid culture was obtained by filling both channels with TrypLE solution in PBS and incubating at 37°C for 20 min or until complete dissociation was achieved using gentle pipetting. The dissociated epithelium was collected from the epithelial channel and digestion was quenched using Advanced DMEM/F-12 (Gibco). All single-cell samples were distributed at $5 \times 10^5$ cells/mL for live staining with mouse anti-human CEA IgG (Santa Cruz) in DPBS with 2% FBS (Sigma). Secondary staining for target was performed using QIFIKIT (BIO-CYTEX) anti-mouse IgG, along with mouse IgG1 Isotype CEA (BioLegend) for secondary control and provided calibration and standard beads. Samples were run with BD FACSCelesta flow cytometer (BD Biosciences), and data analyzed using FlowJo V10 software (FlowJo).

## Live staining and imaging

For timepoints T = 0, 12, 48, and 72 hr after PBMC-TCB administration, effluents were collected for further analysis and Pod inlets were aspirated. CellEvent Caspase-3/7 Green live staining detection reagent (Thermofisher Scientific) at 2 µM was prepared and added to the epithelial and vascular channels in order to visualize an apoptotic T-cell killing response. Pod inlets were aspirated and 300 µL of live staining solution was added to each respective inlet. Chips were flowed at 1000 µL/hr for 10 min to flush, then flow was paused to incubate the stain at 37°C for 30 min. Fresh media was flushed through after incubation and chips were transferred to a confocal laser-scanning microscope (Inverted Zeiss LSM 880, Zeiss) in small groups for live imaging.

## Flow cytometry

PBMC were harvested from Colon Intestine-Chip epithelial channels at the final timepoint after administration. PBMC were washed with DPBS and stained with 2 μM live/dead Fixable Yellow Dead Cell Stain (ThermoFisher) and washed in DPBS. PBMC were then fixed with BD Cytofix (BD Biosciences) fixation solution, washed in DPBS, and either resuspended in 90% FBS (Sigma) + 10% DMSO solution and frozen at −20°C until use or stained immediately. Samples to be stained for surface markers were washed in DPBS and resuspended in Cell Staining Buffer (Biolegend). Surface marker stains were prepared in BD /Cytoperm solution (BD Biosciences) and consisted of anti-human CD3 APC-Cy7 (BioLegend), anti-human CD4 Brilliant Violet 786 (BioLegend), anti-human-CD8-PE/Dazzle-594 (BioLegend), and anti-human CD69 APC (BioLegend).

Sample data was acquired using the BD FACSCelesta flow cytometer (BD Biosciences), and data was analyzed using FlowJo V10 software (FlowJo).

## Immunofluorescence microscopy

Colon and Duodenum Intestine-Chip, Organoid, and Transwell samples were fixed with 4% paraformaldehyde (PFA) (Electron Microscopy Sciences). Samples were then washed twice using DPBS and perfused with a 0.3 M glycine in DPBS (Sigma) solution to remove residual PFA. Chips were cut in half and stored in DPBS and 0.05% sodium azide. Samples were stained overnight at 4°C with the following primary antibodies diluted in CytoPerm/Wash buffer (BD Biosciences): recombinant rabbit anti-CEA (Abcam) for samples without TCB treatment and monoclonal rat anti-CD45 (Invitrogen). After overnight incubation, the chips were washed three times in DPBS, and nuclei were counterstained with DRAQ5 (Thermofisher Scientific) and secondary antibody DyLight 405 AffiniPure Donkey Anti-Rat IgG (H+L) (Jackson ImmunoResearch) diluted in Perm/wash buffer. For samples without TCB treatment, the rabbit anti-CEA was stained with the secondary antibody, donkey anti-rabbit Alexa Fluor-555 (Invitrogen). For samples with TCB treatment, a secondary goat anti-human Alexa Fluor-555 (Invitrogen) was used as the target sites were bound with anti-human TCB after administration. Remaining live imaging signal from CellEvent Detection Reagent was also imaged for all samples.

Confocal laser-scanning microscopy was performed using an Inverted Zeiss LSM 880 (Zeiss). At least three fields of view were taken per chip, from separate random locations along the co-culture channel. Widefield tile images were also acquired on Axio Observer.Z1 (Zeiss) (n=5, per chip co-culture channel).

## Image analysis

Image analysis was performed using the image analysis suite Fiji (National Institute of Health) to quantify PBMC attachment to the Colon Intestine-Chip epithelium. Cell Event Green-positive signal was visualized using confocal laser-scanning microscopy and quantified in Fiji. PBMC attached to epithelium were stained with monoclonal rat anti-CD45 primary antibody, then DyLight 405 AffiniPure Donkey Anti-Rat IgG (H+L) secondary antibody. Tile images acquired at 40x magnification to cover the full microfluidic co-culture channel. Tile images were stitched together through Zen Blue software (Zeiss). Raw images in '.czi' format were converted to '.tiff' format in Fiji software (NIH). Images were processed using a macro script in Fiji which detects PBMC signal that fits a preset size criterion as object 'counts'. Since the PBMC were clustered in-between the epithelial structures this count is described as 'PBMC Clusters' and not individual cells. All image brightness and contrast thresholds were set to the same values for processing, which were determined by using the automatic contrast settings for sample with the highest overall PBMC signal. The size criterion for a PBMC cluster was from 15.9 to 106 $\mu m^2$, which was determined visually.

## Analysis of cytokines

At T=0, 24, 48, and 72 hr after PBMC-TCB administration, effluents were collected from Colon Intestine-Chip Pod inlets and outlets. Effluents were centrifuged to remove debris and then frozen at −20°C until measurement. Measurement of cytokines for Colon Intestine-Chip (IFNɣ, TNFα, Granzyme-B, IL-2, IL-4, and IL-8) was performed using customized Invitrogen ProcartaPlex multiplex immunoassays (reference PPX-12-MXNKRV6). Each kit contained a black 96-well plate (flat bottom plate), antibody-coated beads, detection antibody, streptavidin-R-phycoerythrin (SAPE), reading

buffer, and universal assay buffer. In addition, standards with known concentration were provided to prepare a standard curve. According to the Invitrogen Publication Number MAN0017081 (Revision B.0 (33)), the assay workflow was the following. After adding the beads into the flat bottom plate, the beads were washed using a flat magnet and an automated plate washer (405TS microplate washer from Bioteck). Then standards and samples diluted with a universal buffer were added into the plate and a first incubation started for 2 hr. After a second wash, detection antibodies were added. After 30 min incubation and a wash, SAPE was added. Finally, after 30 min incubation and a last wash, the beads were resuspended in the reading buffer and the plates were ready for analysis.

The data was acquired with a Luminex instrument, BioPlex-200 system from Bio-Rad. Using the Certificate of Analysis provided with the kit, bead region and standard concentration value S1 for each analyte of the current lot were entered in the software, BioPlex Manager. Plotting the expected concentration of the standards against the mean fluorescent intensity (MFI) generated by each standard, the software generated the best curve fit and calculated the concentrations of the unknown samples (in pg/mL). The data were then exported in Excel and plotted in Graphpad Prism.

### Statistics

Experiments were performed with at least triplicates for each chip sample per group. Brightfield images of chips were collected including at least three fields of view per chip at various points throughout the co-culture area of the Intestine-Chips. All graphs are plotted as group means (individual points displayed if n < 10 samples per group) ± SEM. Statistical significance (p<0.05) was determined via one-way or two-way ANOVA using Tukey's multiple comparisons unless specified otherwise.

### Immunohistochemistry human tissues

Immunohistochemical staining for CEA expression in formalin-fixed, paraffin-embedded human intestinal tissues was carried out on a Discovery Ultra automated slide stainer using a rabbit anti-human monoclonal antibody for CEA (Clone T84.66, Roche Glycart AG, Switzerland) at 2.23 µg/mL after antigen retrieval with CC1 (Ventana Medical Systems Inc) on tissues. A secondary antibody was used a donkey anti-rabbit biotinylated polyclonal IgG (Jackson Immoresearch Lab, cat: 711-065-152) at 5 µg/mL, and DAB Map Kit (Ventana 760–124) was used as detection system.

## Acknowledgements

We thank Donald Ingber, Pablo Umaña, Alex Phipps, Thierry Lave, Amy Lambert, Wolfgang Richter, Elisabeth Husar, Ulrike Hopfer, and Lorna Ewart for useful scientific discussions; Nikolai Kaschau for his commitment and guidance on intellectual property; Antonio Varone, Ionnis Moriannis, David Conegliano, and Lian Leng for their contributions to developing the Alveolus Lung-Chip model; Magdalena Kasendra, Raymond Luc, and Athanasia Apostolou for their contributions to developing the Colon Intestine-Chip model; Robin Friedman, Alicia J Stark, Abhishek Shukla, and José Fernandez-Alcon for their contributions to image analysis; Gurpreet Brar for her contributions to flow-cytometry analysis; and Lorna Ewart for her critical review of the manuscript.

## Additional information

### Competing interests

S Jordan Kerns: Is a current employee of and hold equity interests or options to obtain equity interests in (Emulate Inc). Is an inventor on a patent application (P16451EP00/16/912,391) submitted by Hoffmann-LaRoche and Emulate that covers 'Method for Assessing a Compound Interacting with a Target on Epithelial Cells'. Chaitra Belgur, Heather Shannon Grant, Katia Karalis: Is a former employee of and hold equity interests or options to obtain equity interests in (Emulate Inc). Is an inventor on a patent application (P16451EP00/16/912,391) submitted by Hoffmann-LaRoche and Emulate that covers 'Method for Assessing a Compound Interacting with a Target on Epithelial Cells'. Debora Petropolis: Is a former employee of and hold equity interests or options to obtain equity interests in (Emulate Inc). Is an inventor on a patent application (P16451EP00/16/912,391) submitted by Hoffmann-LaRoche and Emulate that covers Method for Assessing a Compound

Interacting with a Target on Epithelial Cells,. Marianne Kanellias: Is a current or former employee of and hold equity interests or options to obtain equity interests in (Emulate Inc). Riccardo Barrile: Is a former employee of and hold equity interests or options to obtain equity interests in (Emulate Inc). Is an inventor on a patent application (P16451EP00/16/912,391) submitted by Hoffmann-LaRoche and Emulate that covers, Method for Assessing a Compound Interacting with a Target on Epithelial Cells'. Johannes Sam, Tina Weinzierl, Tanja Fauti, Anne Freimoser-Grundschober, Jan Eckmann, Martina Geiger, Anneliese Schneider, Anna Maria Giusti, Thomas Singer, Christian Klein, Marina Bacac: Employment, patents and ownership of stock with Roche. Carina Hage, Virginie Micallef, Regine Gerard, Michael Bscheider, Ekaterina Breous-Nystrom: Employment and ownership of stock with Roche. Patrick Ray Ng: is a former employee of and hold equity interests or options to obtain equity interests in (Emulate Inc). William Tien-Street: is a current employee of and hold equity interests or options to obtain equity interests in (Emulate Inc). Dimitris V Manatakis: Is a current employee of and hold equity interests or options to obtain equity interests in (Emulate Inc). Cristina Bertinetti-Lapatki, Adrian B Roth: Is an inventor on a patent application (P16451EP00/16/912,391) submitted by Hoffmann-LaRoche and Emulate that covers 'Method for Assessing a Compound Interacting with a Target on Epithelial Cells'. Employment, patents and ownership of stock with Roche. Geraldine A Hamilton: Is a current or former employee of and hold equity interests or options to obtain equity interests in (Emulate Inc). Is an inventor on a patent application (P16451EP00/16/912,391) submitted by Hoffmann-LaRoche and Emulate that covers 'Method for Assessing a Compound Interacting with a Target on Epithelial Cells'. Annie Moisan: is an inventor on a patent application (P16451EP00/16/912,391) submitted by Hoffmann-LaRoche and Emulate that covers Method for Assessing a Compound Interacting with a Target on Epithelial Cells'. Nikolce Gjorevski, Lauriane Cabon: is an inventor on a patent application (P16451EP00/16/912,391) submitted by Hoffmann-LaRoche and Emulate that covers Method for Assessing a Compound Interacting with a Target on Epithelial Cells'. Employment, patents and ownership of stock with Roche. The other author declares that no competing interests exist.

## Funding

No external funding was received for this work.

## Author contributions

S Jordan Kerns, Formal analysis, Supervision, Validation, Investigation, Visualization, Writing - review and editing; Chaitra Belgur, Software, Formal analysis, Validation, Investigation, Visualization, Writing - review and editing; Debora Petropolis, Data curation, Software, Formal analysis, Supervision, Validation, Investigation, Visualization, Methodology; Marianne Kanellias, Software, Validation, Investigation, Visualization; Riccardo Barrile, Conceptualization, Formal analysis, Supervision, Investigation, Methodology; Johannes Sam, Formal analysis, Visualization, Methodology; Tina Weinzierl, Formal analysis, Supervision, Investigation, Visualization; Tanja Fauti, Formal analysis, Investigation, Visualization; Anne Freimoser-Grundschober, Resources, Supervision, Investigation, Visualization, Methodology; Jan Eckmann, Supervision, Investigation, Visualization, Methodology; Carina Hage, Investigation, Visualization, Methodology; Martina Geiger, Anneliese Schneider, Peter Bruenker, Resources; Patrick Ray Ng, William Tien-Street, Data curation, Investigation; Dimitris V Manatakis, Software, Visualization; Virginie Micallef, Regine Gerard, Investigation, Visualization; Michael Bscheider, Supervision; Ekaterina Breous-Nystrom, Adrian B Roth, Geraldine A Hamilton, Project administration; Anna Maria Giusti, Resources, Investigation, Visualization, Methodology; Cristina Bertinetti-Lapatki, Conceptualization, Supervision; Heather Shannon Grant, Project administration, Writing - review and editing; Thomas Singer, Funding acquisition, Project administration; Katia Karalis, Annie Moisan, Conceptualization, Supervision, Writing - review and editing; Christian Klein, Marina Bacac, Resources, Supervision, Writing - review and editing; Nikolce Gjorevski, Conceptualization, Data curation, Formal analysis, Supervision, Investigation, Visualization, Methodology, Writing - original draft, Writing - review and editing; Lauriane Cabon, Conceptualization, Data curation, Formal analysis, Supervision, Validation, Investigation, Visualization, Methodology, Writing - original draft, Writing - review and editing

## Author ORCIDs

Chaitra Belgur (iD) https://orcid.org/0000-0001-5670-6166
Riccardo Barrile (iD) https://orcid.org/0000-0002-7301-3959
Patrick Ray Ng (iD) https://orcid.org/0000-0003-4566-9173
Nikolce Gjorevski (iD) https://orcid.org/0000-0001-8320-0443
Lauriane Cabon (iD) https://orcid.org/0000-0001-8472-2227

## Ethics

Animal experimentation: The animal facility has been accredited by the Association for Assessment and Accreditation of Laboratory Animal Care (AAALAC). All animal studies were performed in accordance with the Federation for Laboratory Animal Science Associations (FELASA). The animal studies were approved by and done under license from the Government of Upper Bavaria (Regierung von Oberbayern; Approval number: Az 55.2.1.54-2532.0-10-16). We have complied with all relevant ethical guidelines and regulations.

## Decision letter and Author response

Decision letter https://doi.org/10.7554/eLife.67106.sa1
Author response https://doi.org/10.7554/eLife.67106.sa2

# Additional files

## Supplementary files

• Transparent reporting form

## Data availability

RNA sequencing data have been deposited in the National Center for Biotechnology Information Gene Expression Omnibus (GEO) under accession number GSE175821.

The following dataset was generated:

| Author(s) | Year | Dataset title | Dataset URL | Database and Identifier |
|-----------|------|---------------|-------------|-------------------------|
| Kerns SJ, Manatakis DV | 2021 | Genome-wide transcriptome profiling of human organoids, colon intestine-chip and human alveolus-chips using RNA-seq | https://www.ncbi.nlm.nih.gov/geo/query/acc.cgi?acc=GSE175821 | NCBI Gene Expression Omnibus, GSE175821 |

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
