## [Decision Letter]

**Acceptance summary:**

Your manuscript provides interesting findings and a useful platform for exploring safety in developing antibody-based cancer immunotherapy. In this regard and the positive feedback provided by the reviewers, I do feel that it should be published.

**Decision letter after peer review:**

Thank you for resubmitting your work entitled "Human immunocompetent Organs-on-Chips platforms allow safety profiling of tumor-targeted T-cell bispecific antibodies" for further consideration by *eLife*. Your article has been reviewed by 3 peer reviewers, including Ping-Chih Ho as the Reviewing Editor and Reviewer #1, and the evaluation has been overseen by Betty Diamond as Senior Editor.

The manuscript has been improved but there are some remaining issues that need to be addressed, as outlined below:*Reviewer #1:*

Developing platforms for better evaluating actions and safety of immunotherapy is a critical task for translational application; however, this is largely hampered due to the differential expression of target proteins in mouse models and human subjects and the intrinsic differences in T cells from murine and human subjects. This work presents an interesting and highly applicapable platform to address these issues. It represents a major advantage for immuno-oncology field for evaluating drug safety and action kinetics.

This manuscript is clear and straightforward. I do not see obvious weakness and issues. However, I would recommend the authors to take the advantage of this platform to evaluate the following two points.

1. Is it possible to use this platform to depict the working kinetics, including anti-tumor and unwanted killing of normal cells, between antibodies with high and low affinity? In the other word, whether the EC50 calculated by this platform can be close to what observed in mouse model at least?

2. In the in vitro killing of tumor cells, whether this platform can recapitulate the suppressive feature increased in response to the presence of Tregs or MDSCs? Is it possible to introduce this suppressive immune subsets for examining the action of bispecific antibodies?*Reviewer #2:*

This manuscript is of interest to oncologists involved in early drug development of immunotherapy. It very elegantly describes an in vitro platform to study and predict toxicities of new biological therapies. Its main relevance is that it fills a gap between preclinical toxicology studies and first in human clinical trials. The data is presented clearly and the conclusions are consistent with the data.

The authors of this manuscript achieve to demonstrate a very useful in vitro platform for predicting toxicities of biological therapies aimed at human antigens, and where the immune system is activated as a consequence of antigen engagement. Its main relevance resides in that it fills a gap between preclinical toxicology studies and first in human clinical trials.

The major strength is the thoroughness of the data presented. The authors not only use a molecule that is at the preclinical development level, but also one which is already under clinical development. They very elegantly demonstrate that the model can reproduce toxicity observed in humans. The authors achieve their aims with their experiments, and their results support their conclusions.

This in vitro platform will need further study with other targets and biological agents, but the data presented here is already very promising and will help detect immunotherapies too toxic for treating patients, consequently reducing attrition rate of drugs.

*Reviewer #3:*

This paper is of interest to broad readers in the cancer immunotherapy field. It demonstrated that organ-chip system provides improved or otherwise not available human tumor models for immune-oncology studies particularly involving T-cell engaging bispecific antibodies. The key point that these organ chips captures on-target toxicities of such therapies better than other existing models is mostly well supported with few improvements.

In the current manuscript, the authors explored organ chips as a cancer immunotherapy research model to evaluate the safety of tumor antigen-targeting T-cell engaging bispecific antibodies. Using two independent human organ chip systems (lung and intestine), and tumor antigens (FOLR1 and CEA), the authors demonstrated that organ chips enabled toxicity evaluation of T-cell engaging bispecific antibodies (TCBs) that would otherwise not be possible in xenograft or non-human primate model systems. The manuscript also sought to compare the organ chip system to the other ex vivo model systems such as 2D transwell and 3D organoid cultures. The main conclusions of this paper are supported while some experiments need to be clarified, extended or discussed.

The comparison of average FOLR1 expression levels between alveolar chips and Hela cells might be misleading due to the difference in heterogeneity of the primary cells and cell line. As shown in Figure 1I and Figure S3A, only a subset of alveolar cells express FOLR1 while Hela cells uniformly express high levels of FOLR1. However, in Figure 1E healthy human lung IHC, FOLR1 expression seemed quite uniform along the alveoli. Can the authors provide an explanation for this difference between the chip and native tissues? Also, what's the molecular density of FOLR1 on the FOLR1+ chip cells? Is it still lower than Hela or at similar levels?

Analysis for the immune cells needs to be refined for the current study.

1. The flow cytometry analysis of the merely CD69+CD8^+^% is very limited to evaluate the true effects of these TCBs on seeded PBMCs. More analysis of the T cell populations should be presented to give the readers a more complete pictures of the T cell activation in the chips. For example, was the % of the total T/CD4/CD8 populations increased in PBMCs recovered from the chips treated with TCBs? Or was merely the activation status alternated.

2. It's hard to tell in Figure 2A this type of imaging analysis whether the Casp3/7+ cells are epithelial cells or PBMCs. An orthogonal approach to evaluate the actual killing (live versus dead cells) of epithelial cells upon TCB treatment is highly recommended to support the conclusion.

3. Are the PBMC attached to the epithelium T cells? A co-staining with CD3 can be very helpful.

The authors made a solid point that toxicity against FOLR1+ alveolar epithelial cells could be lowered by using a FOLR1(LO) TCB. What the authors haven't shown as the CEA case is whether this low affinity antibody still possesses sufficient in vivo tumor control efficacy?

4. As the PBMCs introduced in the "chip" haven't been exposed to the tumor micro-environment the authors should tempered down their conclusions. Therefore, the authors should either expose the immune cells/PBMCs into the tumor condition media or introduce tumor cells into the chip or dedicate a paragraph in the Discussion section in order to provide more clarity of how their conclusions may or not differ in the presence of the TME. In addition, the level of T cell activation/exhaustion will be different in the presence of the TME. It would be interesting to examine the effect of the TMB across T cells with different activation status (naïve, activated/effector, exhausted etc). All the above should be discussed in the Discussion section of the revised manuscript.

5. The authors should provide all the in vitro evidence on how they validated the control (non-tumor targeted) TCBs.

6. The authors described that a 48h incubation time is sufficient to capture changes in the cytokine secretion profile. Indeed, their data do support this finding. Is this the optimal time point and would this time point be optional for all TCBs or conventional bivalent antibodies? Have the authors performed a time course experiment? The reason because several discrepancies have been described in the literature where a super-agonist CD28 antibody in cynomolgus monkeys failed to predict cytokine release syndrome (CRS) at 24 and 48h, and its subsequent catastrophic effect in healthy individuals. Others, have shown that after CD28 super-agonist administration in cynomolgus monkeys, cytokines were detected at earlier times points (5h) could detect cytokines production. The authors should discuss how these findings may affect their conclusions/thinking process. Have you ever tested a CD28 superagonist antibody in this system?

---

## [Author Response]

Reviewer #1:This manuscript is clear and straightforward. I do not see obvious weakness and issues. However, I would recommend the authors to take the advantage of this platform to evaluate the following two points.1. Is it possible to use this platform to depict the working kinetics, including anti-tumor and unwanted killing of normal cells, between antibodies with high and low affinity? In the other word, whether the EC50 calculated by this platform can be close to what observed in mouse model at least?

For the unwanted killing of normal cells, we could not use mouse models as toxicities were not observed in these cases (neither for FOLR1 nor CEA targeting antigens). In cynomolgus studies, FOLR1(Hi) pulmonary toxicity was observed at 10 µg/kg single i.v. injection within 24 hours after injection and the Cmax at that dose was calculated to be 0.34 µg/mL. We observed that FOLR1(Hi) showed toxicity in the Alveolus Lung-chip starting from 0.2 µg/mL and 24h time point whereas it took 20 µg/mL for the FOLR1(Lo) antibody to induce alveolar epithelial cell killing. The 0.2 µg/mL concentration is very close to the 0.33 µg/mL measured as an in vivo blood concentration in the preclinical cynomolgus study but should however be taken with a lot of caution since we do not know the antibody concentration reached in the local pulmonary alveolar region upon treatment in animals and in humans.

For the anti-tumor activity of the antibodies, several in vitro tumor cell lines experiments and in vivo mouse experiments have been performed for both targets.

For FOLR1 (Hi) TCB, we compared the EC50 obtained with standard in vitro killing experiments (as presented in Figure 1- supplement figure 1F) to the EC50 obtained with chip experiments (such as the one presented in Figure 2- supplement figure 2). The EC50 in standard 2D HeLa assay was 2.2 pM whereas the EC50 in chips was estimated at 1.1 pM, overall in a similar dose range. To provide that information to the reader, we added a sentence (lines 197-198) stating “Of note, FOLR1(Hi) TCB EC50 was estimated at 1.1 pM which was close to the value obtained from standard 2D in vitro killing experiments (2.2 pM).” Despite a lower toxicity towards healthy alveolar epithelial cells, FOLR1(Lo) TCB could still efficiently kill HeLa cells due to the high amount of accessible FOLR1 molecules at their cell surface. Indeed the calculated EC50 in standard 2D HeLa killing assays was 1.3 pM, so very close to FOLR1(Hi) TCB EC50 values.

EC50 were not calculated from mouse experiments due to the fact that this wide range of doses is not commonly run for these bispecific molecules. We did however perform mouse efficacy studies comparing both FOLR1(Hi) and (Lo) TCBs in a HeLa transfer model in humanized PBMC engrafted NOG mice. We represented tumor weights at termination of animals for all treatment conditions. FOLR1(Lo) TCB was administered at two different doses, 0.5 mg/kg once a week and 2.5 mg/kg once a week. At both doses, the reduction in tumor weight was significantly lower than the vehicle control group and comparable to the one obtained with FOLR1(Hi) TCB, corroborating the hypothesis that the lower affinity variant retained efficacy against high FOLR1 expressing tumor cells despite a lower toxicity pulmonary risk. Thanks to our reviewer’s comment, we added the in vivo data mentioned above in the revised manuscript as an additional reference in Figure 1- supplement figure 1G.

2. In the in vitro killing of tumor cells, whether this platform can recapitulate the suppressive feature increased in response to the presence of Tregs or MDSCs? Is it possible to introduce this suppressive immune subsets for examining the action of bispecific antibodies?

We believe the platform could be used for that purpose as it is technically possible to add either circulating Tregs in the bottom/top channel or resident myeloid-derived suppressor cells embedded in the matrix below the pulmonary or intestinal epithelium. Of note, Tregs present in the PBMC introduced in the apical channel of HeLa chips (Figure 2- supplement figure 2) did not prevent the killing of target cells by T cells in presence of FOLR1(Hi) TCB although it may have reduced or delayed it.

Given that the primary goal of our manuscript is to demonstrate the utility of the platform for safety assessment of on-target off-tumor crossreactivity of T cell engagers, we however feel that investigating these highly relevant questions lie more within the scope of a manuscript directed towards the tailoring of the platform for efficacy questions. Therefore we hope our reviewer respects that we did not run additional experiments but instead mention this possibility in our discussion.

Reviewer #3:The comparison of average FOLR1 expression levels between alveolar chips and Hela cells might be misleading due to the difference in heterogeneity of the primary cells and cell line. As shown in Figure 1I and Figure S3A, only a subset of alveolar cells express FOLR1 while Hela cells uniformly express high levels of FOLR1. However, in Figure 1E healthy human lung IHC, FOLR1 expression seemed quite uniform along the alveoli. Can the authors provide an explanation for this difference between the chip and native tissues? Also, what's the molecular density of FOLR1 on the FOLR1+ chip cells? Is it still lower than Hela or at similar levels?

Figure 1E shows an area with strong diffuse staining by IHC for FOLR1 in a healthy human lung. It was chosen on purpose, in order to allow the reader to identify the positive staining, which is not easily appreciated because of the very thin structure of alveolar epithelial cells. Other areas of the same sample and other human lung samples also showed areas with less abundant or less uniform staining (see Author response image 1; two representative pictures obtained in a healthy human lung tissue where this variation of staining is clearer).

**Author response image 1. sa2fig1:** 

We also observed the difference in expression in the HeLa cells prepared for IHC. Indeed, as visible in Figure 1D, some regions are less positive than others for FOLR1 staining (right area of the selected picture). That difference may be due to upregulation of the *FOLR1* gene in response to cellular growth or in response to folate levels (Doucette and Stevens, 2001; Sabharanjak and Mayor, 2014).To complement the IHC approach, the molecular density of FOLR1 on the alveolus chip cells was evaluated with the QIFIKIT method which is intended for the quantitative determination of cell surface antigens by flow cytometry using indirect immunofluorescence assay. Given the mode of action of T cell bispecifics tested in the study, accessible surface FOLR1 is the fraction of the receptor expected to drive any toxicity or efficacy. QIFIKIT consists of a series of 6 bead populations coated with different, but well-defined quantities of a mouse monoclonal antibody. The beads mimic cells labeled with a specific primary mouse monoclonal antibody. The results we obtained are presented in Figure 1J for alveolar cells and in Figure 2-supplement figure 2 for HeLa cells comparison. The two sentences lines 132-137 state that finding “Flow cytometry-mediated quantification allowed us to estimate the cell surface expression of FOLR1 within the Alveolus Lung-Chip at an average of ~10,000 molecules expressed per cell (Figure 1J). For comparison, the high FOLR1 expressing ovarian carcinoma HeLa cell line displayed an average of ~450 000 molecules per cell when cultured on chip (Figure 2-supplement figure 2), confirming the difference observed in IHC between healthy and tumor cells.” The calculation shows the average amount of FOLR1 binding sites per cell in the analyzed population. Because it is a mean value, it does not illustrate the heterogeneity of the epithelium observed by immunofluorescence. Therefore we added Author response image 2, a representative graph of the QIFIKIT flow cytometry panel which shows that the FOLR1 cell surface staining by FACS does not identify two populations (one FOLR1lo and one FOLR1hi) but rather a continuum of cells in a similar log detection scale (blue population). For comparison we added HeLa cells dot plots as positive controls (pink population). Indeed, we found that HeLa cells expressed a lot more surface FOLR1 than any highly expressing FOLR1+ primary alveolar epithelial cell measured across the different donors tested on chips.

Analysis for the immune cells needs to be refined for the current study.1. The flow cytometry analysis of the merely CD69+CD8^+^% is very limited to evaluate the true effects of these TCBs on seeded PBMCs. More analysis of the T cell populations should be presented to give the readers a more complete pictures of the T cell activation in the chips. For example, was the % of the total T/CD4/CD8 populations increased in PBMCs recovered from the chips treated with TCBs? Or was merely the activation status alternated.

We complemented the main figures with a supplementary figure (Figure 3- supplement figure 1) which compiled a more detailed analysis of the subsets of immune cells harvested from the alveolus lung chips.

We observed that there was no modification of % total T, CD4 and CD8 populations across control and treated conditions but that FOLR1 TCBs induced a change in the CD69^+^ activation status of both CD4 and CD8 T cells.

2. It's hard to tell in Figure 2A this type of imaging analysis whether the Casp3/7+ cells are epithelial cells or PBMCs. An orthogonal approach to evaluate the actual killing (live versus dead cells) of epithelial cells upon TCB treatment is highly recommended to support the conclusion.

We agree and the caspase 3/7^+^ epithelial cells had to be discriminated from the PBMCs indeed. For that purpose we either used the fluorescent CellTracker signal for the alveolus lung chip or performed an anti-CD45 additional staining for the intestine chip. For both lung and intestine chips, we updated the methods section with a more detailed description of the image analysis workflow applied to precisely quantify apoptotic epithelial cells and attached PBMC (see line 687 and line 955).

3. Are the PBMC attached to the epithelium T cells? A co-staining with CD3 can be very helpful.

To answer that question we performed an additional immunostaining of the chips with an anti-CD3 antibody as suggested. We then quantified the amount of T cells attached to the epithelium and percentage of T cells among CellTracker positive attached PBMC. Based on these results, we conclude that a higher amount of T cells attached to the epithelium in presence of FOLR1(Hi) TCB but that population was not representing a higher proportion of PBMC in comparison to the control conditions. This finding suggests that non CD3 positive immune cells (such as monocytes) also attach to the epithelium probably as a consequence of their own activation as reflected by the nature of cytokines released. These results have been added to figure 2F.

The authors made a solid point that toxicity against FOLR1+ alveolar epithelial cells could be lowered by using a FOLR1(LO) TCB. What the authors haven't shown as the CEA case is whether this low affinity antibody still possesses sufficient in vivo tumor control efficacy?

We agree this dataset was missing from our Figure 1- supplement figure 1 and we now added in vivo experiments performed in a HeLa/ human PBMC engrafted NOG model (see figure in the revised manuscript). Interestingly, the lower affinity variant of FOLR1 TCB was similarly efficacious despite a reduced toxicity risk. We explained this finding by the high expression of FOLR1 by HeLa cells which is sufficient for FOLR1(Lo) TCB to be as efficient as the higher affinity variant (also seen in vitro with calculated EC50 of 1.3pM and 2.2pM respectively). On the other hand, a low expression of FOLR1 on alveolar epithelial cells was not sufficient for FOLR1(Lo) TCB to elicit killing at low doses like FOLR1(Hi) TCB.

4. As the PBMCs introduced in the "chip" haven't been exposed to the tumor micro-environment the authors should tempered down their conclusions. Therefore, the authors should either expose the immune cells/PBMCs into the tumor condition media or introduce tumor cells into the chip or dedicate a paragraph in the Discussion section in order to provide more clarity of how their conclusions may or not differ in the presence of the TME. In addition, the level of T cell activation/exhaustion will be different in the presence of the TME. It would be interesting to examine the effect of the TMB across T cells with different activation status (naïve, activated/effector, exhausted etc). All the above should be discussed in the Discussion section of the revised manuscript.

We thank our reviewer for the interesting and important suggestion. While we believe that using naive PBMCs for safety evaluation is a reasonable approach in the case when the organ of concern is different from the cancer-affected organ, we fully agree that TME effects and anti-tumor efficacy effects should be taken into account when the target-expressing organ and the tumor co-localize. For example, treating colorectal cancer with CEA-targeting TCBs may lead to an increased risk of intestinal toxicities, bearing in mind that tumor cell lysis may lead to the release of inflammatory cytokines and tumor antigens, thus further potentiating immune cell activation and cytotoxicity.

We followed the reviewer’s suggestion and performed an experiment wherein PBMCs were exposed to the CEA-expressing SNU1544 colon adenocarcinoma cell line, which leads to tumor cell lysis and cytokine release in the presence of CEA(Hi) TCB. Following 24 h exposure, the tumor cell-primed PBMC were introduced into the Colon Intestine-Chip along with CEA(Hi) TCB and the safety effects evaluated by monitoring cytokine release. Surprisingly, we observed no significant difference in the levels of cytokines released by naive PBMCs and those pre-exposed to tumor cells in the presence of TCB (please see Author response image 3). We would refrain from drawing strong conclusions based on these data, however, owing to several important experimental caveats. First, the system comprised a cancer cell line in 2D culture, which is a far cry from the tumor microenvironment. Second, based on the TCB mode of action, the most extensively activated T cells would be those physically crosslinked to the tumor cells during the pre-incubation step, which in turn might prevent their efficient harvesting and transfer to the Intestine-Chip. In other words, the most potent T cells might be left behind. Owing to these limitations, we decided to not include these data in the revised manuscript, see Author response image 3, for the reviewers only.

We have now included a paragraph in the Discussion section addressing the reviewers points. Specifically, we have communicated that the presence of a tumor in the organ with safety liabilities adds complexity to the question. We believe that aspects of this situation could be captured in vitro by introducing a tumor-like compartment within the device, via, for example, tumor organoids.

**Author response image 3. sa2fig3:** 

5. The authors should provide all the in vitro evidence on how they validated the control (non-tumor targeted) TCBs.

In our method paragraph (lines 477-480), we have now added references of previously published manuscripts in which the control non tumor targeted TCBs are described to be inert in Jurkat NFAT reporter assay (for quantification of CD3 downstream signaling) and in various cell lines cytotoxic assays (for non tumor targeting assessment).

6. The authors described that a 48h incubation time is sufficient to capture changes in the cytokine secretion profile. Indeed, their data do support this finding. Is this the optimal time point and would this time point be optional for all TCBs or conventional bivalent antibodies? Have the authors performed a time course experiment? The reason because several discrepancies have been described in the literature where a super-agonist CD28 antibody in cynomolgus monkeys failed to predict cytokine release syndrome (CRS) at 24 and 48h, and its subsequent catastrophic effect in healthy individuals. Others, have shown that after CD28 super-agonist administration in cynomolgus monkeys, cytokines were detected at earlier times points (5h) could detect cytokines production. The authors should discuss how these findings may affect their conclusions/thinking process. Have you ever tested a CD28 superagonist antibody in this system?

We observed that the timing may vary from one organ system to another and would therefore recommend performing a time course experiment for each class of molecule to be tested in this platform. Interestingly the platform is fully compatible with sequential sampling of the outlet channels and further cytokines detection and we took advantage of that feature for the lung alveolus-chip experiment for instance. We would like to precise that although 48h was sufficient to detect toxicities with the lung alveolus-chip, we needed 72h for the colon-chip to detect the maximum overall cytokine release upon TCB treatment (please see Author response image 4). Also we observed that certain cytokines that are released earlier in the cascade, for instance TNFɑ, were better captured at 24h than 48h (see heatmap on Figure 3D).

Regarding the testing of a CD28 super-agonist in the described platform we have not tested it. Although the technology may be biologically relevant to CD28 biology, our manuscript focused on off-tumor on-target liabilities of T cell engagers that today cannot be easily recapitulated in simpler assays. Tegenero effects have been recapitulated (at least to some extent) in high density PBMC culture assays for instance (Hünig T. Nat Rev Immunol. 2012 Apr 10;12(5):317-8.) and may not require a Organ-on-chips platform for its biology. We however think the platform could be of interest to the field and therefore mention it in our discussion (line 420) as an additional class of molecules to be tested similarly to CAR-T cells and checkpoint inhibitors.

**Author response image 4. sa2fig4:**